# SARS-CoV-2 inhibits induction of the MHC class I pathway by targeting the STAT1-IRF1-NLRC5 axis

Ji-Seung Yoo [1], Michihito Sasaki [2], Steven X. Cho[1], Yusuke Kasuga [1], Baohui Zhu [1], Ryota Ouda[1], Yasuko Orba [2,3], Paul de Figueiredo[4,5], Hirofumi Sawa [2,3,6] & Koichi S. Kobayashi [1,5✉]

The MHC class I-mediated antigen presentation pathway plays a critical role in antiviral immunity. Here we show that the MHC class I pathway is targeted by SARS-CoV-2. Analysis of the gene expression profile from COVID-19 patients as well as SARS-CoV-2 infected epithelial cell lines reveals that the induction of the MHC class I pathway is inhibited by SARS-CoV-2 infection. We show that NLRC5, an MHC class I transactivator, is suppressed both transcriptionally and functionally by the SARS-CoV-2 ORF6 protein, providing a mechanistic link. SARS-CoV-2 ORF6 hampers type II interferon-mediated STAT1 signaling, resulting in diminished upregulation of NLRC5 and IRF1 gene expression. Moreover, SARS-CoV-2 ORF6 inhibits NLRC5 function via blocking karyopherin complex-dependent nuclear import of NLRC5. Collectively, our study uncovers an immune evasion mechanism of SARS-CoV-2 that targets the function of key MHC class I transcriptional regulators, STAT1-IRF1-NLRC5.

[1] Department of Immunology, Hokkaido University Graduate School of Medicine, Sapporo 060-8638, Japan. [2] Division of Molecular Pathobiology, International Institute for Zoonosis Control, Hokkaido University, Sapporo 001-0020, Japan. [3] International Collaboration Unit, International Institute for Zoonosis Control, Hokkaido University, Sapporo 001-0020, Japan. [4] Department of Veterinary Pathobiology, Texas A&M University, College Station, TX 77843, USA. [5] Department of Microbial Pathogenesis and Immunology, Texas A&M Health Science Center, Bryan, TX 77807, USA. [6] One Health Research Center, Hokkaido University, Sapporo 001-0020, Japan. ✉email: kskobayashi@med.hokudai.ac.jp

Coronavirus disease-2019 (COVID-19) caused by severe acute respiratory syndrome coronavirus (SARS-CoV)-2 has become a global emergency. SARS-CoV-2 is an emerging zoonotic pathogen belonging to *Betacoronavirus* that infects human epithelial cells mainly in the respiratory tract and intestine through interactions with angiotensin-converting enzyme 2 (ACE2). The rapid spread of COVID-19 has resulted in health and economic crises in many countries. Recent studies have suggested that SARS-CoV-2 possesses multiple immune evasion strategies, resulting in impaired host antiviral responses[1]. COVID-19 patients demonstrate a tendency towards reduced quantity and quality of functional T lymphocytes, which is closely associated with disease severity and high mortality[2–6]. However, the underlying mechanism by which SARS-CoV-2 inhibits host antiviral programs to eliminate virus-infected cells is poorly understood.

The major histocompatibility complex (MHC) class I-mediated antigen presentation pathway plays an essential role in host defense against intracellular pathogens and cancer. MHC class I contributes towards antiviral immunity by facilitating the presentation of viral antigens to CD8 cytotoxic T cells. Consequently, activated CD8 cytotoxic T cells specifically eliminate virus-infected cells[7]. The type II interferon (IFN) system plays an essential role in the activation of the MHC class I pathway. IFN gamma (IFNγ) stimulation can rapidly activate the transcription factor, STAT1, and subsequently induce IRF1 to upregulate the expression of genes that are essential components of the MHC class I pathway[8,9]. Type I IFNs such as IFN alpha and beta induce similar transcriptional machinery, although less potently[10,11]. In addition to the involvement of type II IFN-induced transcription factors, activation of MHC class I requires a transcriptional co-activator that orchestrates the function of the transcriptional and epigenetic factors on the MHC class I promoter. We have identified a NOD-like receptor family CARD domain containing five (NLRC5) as an MHC class I transactivator (CITA). NLRC5 serves as a master co-activator of MHC class I genes and is critical for the expression of not only MHC class I genes but also essential components of the MHC class I pathway[12,13]. Active NLRC5 undergoes a possible conformational change in the presence of ATP, allowing for nuclear translocation and function as a CITA. The bipartite nuclear localization signal (NLS) in NLRC5 allows NLRC5 to translocate into the nucleus. Although NLRC5 itself does not carry a DNA binding domain, NLRC5 associates with key transcription factors on the MHC class I promoter, generating an active protein/DNA complex termed the CITA enhanceosome. Studies using *Nlrc5*-deficient mice demonstrated that NLRC5 plays a critical role in CD8 cytotoxic T cell activation and protection against infection of intracellular pathogens such as *Listeria monocytogenes* or influenza A virus[14,15].

As MHC class I-dependent immune responses provide key host defense mechanisms against viral infection, many viral species have evolved diverse strategies for targeting the MHC class I pathway to evade host immunity. For example, Herpesviridae family viruses such as Epstein-Barr virus (EBV)[16], Kaposi's sarcoma-associated herpesvirus (KSHV)[17], Herpes simplex virus (HSV)[18], and Human cytomegalovirus (HCMV)[19] are known to inhibit the function of multiple components of the MHC class I pathway. Moreover, human immunodeficiency virus (HIV)-1 Nef[20], murine CMV (MCMV) gp48[21], and Adenovirus E1A can downregulate the expression of MHC class I molecules[22]. Recent studies also showed that Middle East respiratory syndrome (MERS)-CoV and H5N1 influenza suppress MHC class I gene expression by modulating epigenetic regulation of the MHC class I gene locus[23]. However, whether SARS-CoV-2 modulates the MHC class I pathway is poorly understood.

In this study, we report that the MHC class I pathway is impaired by SARS-CoV-2 infection. Induction of MHC class I genes is inhibited in the airway epithelial cells of COVID-19 patients, as well as in the epithelial cell lines infected with SARS-CoV-2. We found that both transcriptional upregulation and functional CITA activity of NLRC5 are suppressed by the ORF6 protein of SARS-CoV-2, which exerts its suppressive activity by inhibiting the interferon-mediated signaling as well as karyopherin complex-dependent nuclear import. Our discovery reveals an immune evasion mechanism by SARS-CoV-2 to escape the MHC class I pathway and may provide a potential therapeutic intervention for COVID-19.

## Results

**Upregulation of the MHC class I genes is suppressed during SARS-CoV-2 infection.** To assess the effect of SARS-CoV-2 infection on the MHC class I pathway, we analyzed an RNA-seq dataset of nasopharyngeal swabs from SARS-CoV-2 infected or non-infected human patients[24] (GSE152075). Given the inherent variability in host response and sampling time of patients being swab tested, we reasoned that mRNA from immune cell influx to the sampled area could mask epithelial cell gene response to infection. Indeed, we observed a strong positive correlation between the pan-immune cell marker, CD45 (gene name *PTPRC*), and MHC class I genes (Supplementary Fig. 1a). Moreover, our analysis of single-cell RNA-seq data from normal human lung[25] demonstrated the predominant source of MHC class I gene expression in the lung to be immune cells (Supplementary Fig. 1b). To minimize gene expression differences potentially influenced by immune cell influx, we restricted our analysis to only include SARS-CoV-2 infected patients who had *PTPRC* expression in a similar range to non-infected patients ("Methods").

We observed a subset of SARS-CoV-2 infected patients to have reduced expression of MHC class I genes compared to non-infected patients (Fig. 1a). Overall, the SARS-CoV-2 infected group had a higher median age (55 vs 48 years old) compared to non-infected patients (Fig. 1b), with no significant difference in gender (43% male vs. 45% male). Despite the increased *PTPRC* expression in the SARS-CoV-2 infected group (Fig. 1b), there was an up to 66% reduction in mean expression of MHC class I genes *HLA-A*, *HLA-B*, *HLA-C*, *B2M*, and *PSMB9*, with no significant change to *NLRC5* or *TAP1* (Fig. 1c). Our analysis of an independent RNA-seq dataset of SARS-CoV-2 infected primary human bronchial epithelial cells[26] corroborates our findings to demonstrate a SARS-CoV-2 specific lack of MHC class I gene induction as opposed to the up to 20-fold increased MHC class I-related gene induction by the innate immunity-triggering influenza A virus mutant strain, IAVΔNS1, or IFNβ stimulation (Fig. 1d). Moreover, MHC class I and related genes were not induced in ex vivo SARS-CoV-2 infected bronchial ciliated cells compared to non-infected bystander cells[27] (Supplementary Fig. 2), further corroborating that SARS-CoV-2 possesses an immune evasion strategy targeting the MHC class I pathway.

To confirm epithelial cell-specific inhibition of upregulation of the MHC class I genes by SARS-CoV-2, we performed infection time-course experiments using three different epithelial cell lines, Calu-3, Caco-2, and Huh7, corresponding to the lung, colon, and liver, respectively (Fig. 2a and Supplementary Fig. 3). qPCR analysis demonstrates that while poly(I:C)-stimulated controls showed significant induction of MHC class I pathway-related gene expression, inhibition of upregulation of the MHC class I-related genes was found in SARS-CoV-2 infected cells (Fig. 2a and Supplementary Fig. 3). Consistent with data from SARS-CoV-2 infected human specimens (Fig. 1), the expression of the primary transcriptional regulators responsible for the MHC class I activation, such as *NLRC5*, *IRF1*, and *STAT1* was not induced

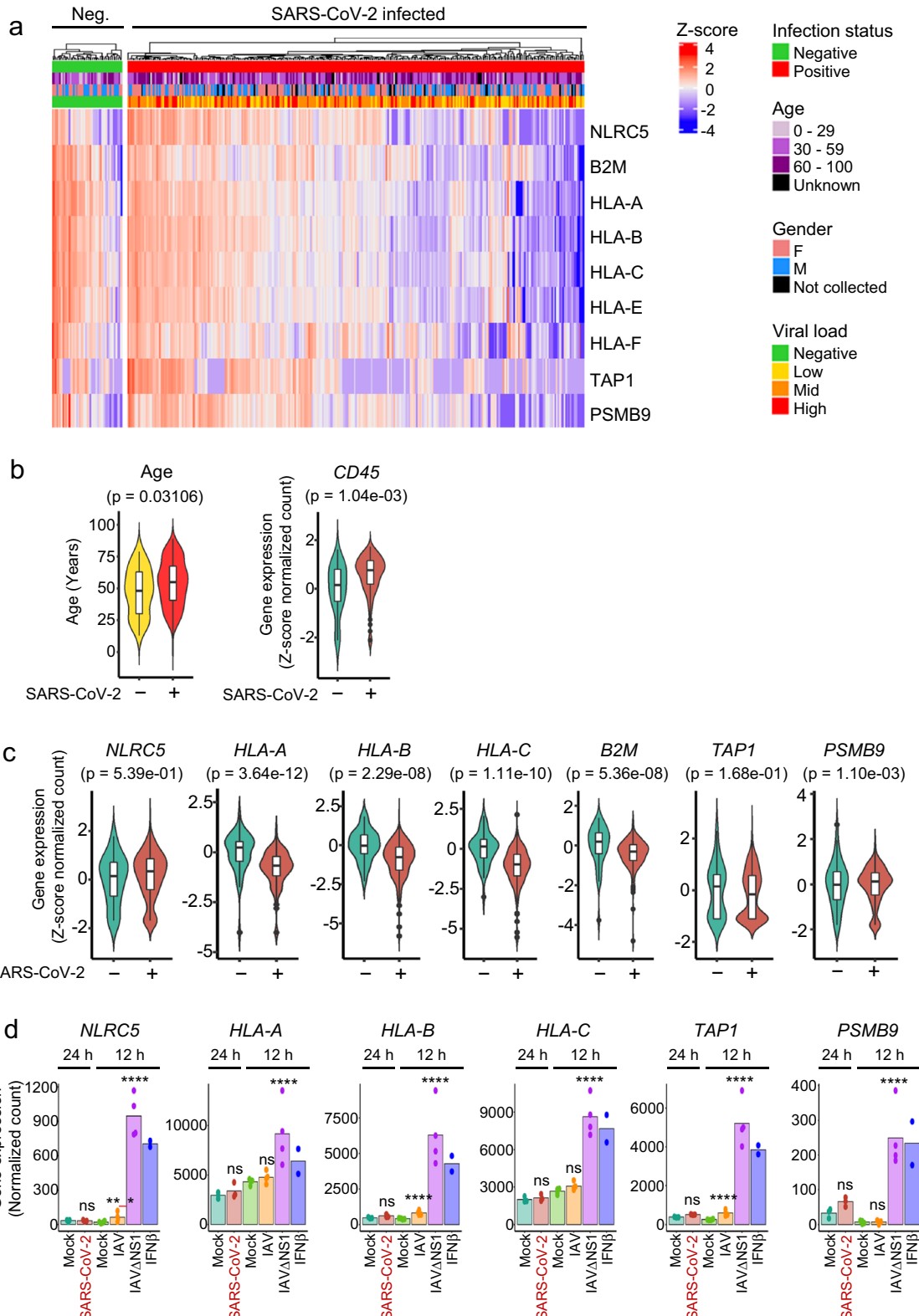

(Calu-3 and Huh7) or induced at an only modest level (Caco-2) upon SARS-CoV-2 infection (Fig. 2b and Supplementary Fig. 3). To evaluate the level of the MHC class I surface expression by SARS-CoV-2 infection, we performed flow cytometric analysis with Caco-2 and HEK293T stably expressing ACE2 (ACE2-HEK293T) cells using various immunostimulants, such as Zika virus (ZIKV) and Tula virus (TULV), poly(I:C), 5' triphosphate-

containing RNA (5'-ppp RNA)[28], and IFNγ. We chose ZIKV (*Flaviviridae*) and TULV (*Hantaviridae*) because these viruses are known to inhibit the interferon signaling pathways similar to SARS-CoV-2 but can activate the MHC class I pathway[29–31]. Thus, these viruses can be a useful control to scrutinize the inhibitory effect of SARS-CoV-2 on MHC class I expression under the condition of minimal IFN effects. Although stimulation

**Fig. 1 Induction of MHC class I genes is suppressed during SARS-CoV-2 infection.** RNA-seq of nasopharyngeal swabs from SARS-CoV-2 negative ($n = 46$) or positive ($n = 300$) patients were controlled for similar expression of CD45 and evaluated for **a** MHC class I gene expression depicted by heatmap of z-score transformed log2 normalized count. **b** Age and CD45 comparisons between SARS-CoV-2 negative and positive patients from **a**. Boxplot center indicate medians with the lower and upper bounds of the box indicating the 25th and 75th percentiles. The upper and lower whiskers extend out 1.5 *IQR from the respective upper and lower bounds on the box. Two-sided Mann–Whitney $U$ test $P$-values are depicted. **c** Violin plots for the expression of indicated MHC class I and related genes, comparing SARS-CoV-2 negative or low immune influx SARS-CoV-2 positive patients from **a**. Graphs show z-score transformed log2 normalized count compared to the SARS-CoV-2 negative group. Boxplots indicate medians ± IQR. Benjamini–Hochberg adjusted two-sided Wald-test $p$-values are depicted. **d** Normal human bronchial epithelial cells were infected with SARS-CoV-2 at a moi of 2 for 24 h or influenza A virus (IAV) or IAV lacking the NS1 gene (IAVΔNS1) at a moi of 3 for 12 h or 100 U/mL IFNβ for 12 h and evaluated for MHC class I gene expression by RNA-sequencing. Bars depict means. Benjamini–Hochberg adjusted two-sided Wald-test $P$-value: **$P < 0.01$; ***$P < 0.001$; ****$P < 0.0001$. $n = 7$ for mock, 3 for SARS-CoV-2, 4 for both IAV and IAVΔNS1, and 2 for IFNβ.

with ZIKV, TULV, poly(I:C), 5'-ppp RNA, and IFNγ could induce the surface expression of the MHC class I molecules, SARS-CoV-2 infected Caco-2 and ACE2-HEK293T showed little or no induction of HLA surface expression (Fig. 2c and Supplementary Figs. 4 and 5). There is an inconsistency of the inhibitory effect between mRNA (Fig. 2a and Supplementary Fig. 3; reduced) and protein (Fig. 2c and Supplementary Figs. 4 and 5; little or no change) of the MHC class I-related genes. Presumably, these inconsistencies may consist of differences in the regulation of gene transcription and surface protein levels, which involve differential protein turnover rates. Nevertheless, upregulation of MHC class I was not observed in any cell types we analyzed at both mRNA and protein levels, indicating that SARS-CoV-2 infection inhibits upregulation of the MHC class I pathway. Collectively, these results demonstrate that targeting of the MHC class I pathway by SARS-CoV-2 may be a potential immune evasion mechanism in epithelial cells.

**SARS-CoV-2 ORF6 suppresses the type II IFN signaling pathway.** Recent studies have reported that SARS-CoV-2 viral proteins can inhibit type I and III IFN signaling pathways by various mechanisms[26,32,33]. We confirmed that compared to poly I:C stimulation, SARS-CoV-2 infection could strongly suppress the host antiviral response, including both viral sensing (represented by TBK1 phosphorylation) and IFN signaling pathways (represented by STAT1 phosphorylation) (Fig. 3a), with little or delayed expression of antiviral innate immune response genes (Fig. 3b). We also observed that ZIKV and TULV virus infection showed little or no induction of the antiviral innate immune responses, as reported previously (Fig. 3b, and Supplementary Fig. 6). However, research related to the suppression of type II IFN signaling by SARS-CoV-2 has not been reported. Given that the type II IFN system is a primary regulator of MHC class I pathway activation, we reasoned that the IFNγ-induced downstream signal cascade might be targeted by SARS-CoV-2 viral protein(s). Thus, we monitored whether SARS-CoV-2 viral protein can inhibit an IFNγ-induced host immune response. We tested five SARS-CoV-2 viral genes, Nsp1, Nsp15, ORF6, ORF8, and N, that have been previously reported to antagonize the host innate immune responses, including type I IFN (Nsp1, ORF6, N)[32–34], viral RNA sensing (Nsp15)[35], and antigen-presenting pathways (ORF8)[36]. While these viral proteins did not prevent IFNγ-induced STAT1 phosphorylation (Fig. 3c), the nuclear import of STAT1 induced by IFNγ stimulation was strongly inhibited by ORF6 (Fig. 3d). These results are consistent with recently published studies that have shown that ORF6 can inhibit type I IFN-mediated STAT1 nuclear translocation[32,33]. As the STAT1 promoter has a gamma interferon-activated sequence (GAS) element[37,38], the expression of STAT1 can be induced by activated STAT1 upon IFNγ stimulation[39]. Therefore, we performed a time-course experiment to examine whether ORF6 can suppress STAT1 expression upon IFNγ stimulation. As expected,

IFNγ-mediated induction of STAT1 mRNA (Fig. 3e) and protein (Fig. 3f) expression was attenuated in the presence of ORF6, while other SARS-CoV-2 proteins we tested did not exhibit inhibitory effects on IFNγ-mediated STAT1 expression (Supplementary Fig. 7). Overall, these results suggest that SARS-CoV-2 ORF6 antagonizes type II IFN-mediated host immune responses by inhibiting STAT1 nuclear translocation.

**SARS-CoV-2 ORF6 inhibits IRF1 and NLRC5 expression as well as IRF1 function.** IRF1 and NLRC5 are two key transcriptional regulators for MHC class I activation, and their expression is driven by the IFNγ-STAT1 signaling axis[12]. Because the induction of the gene expression of IRF1 and NLRC5 was suppressed in SARS-CoV-2 infected epithelial cells (Fig. 2 and Supplementary Figs. 4 and 5), we screened the specific viral antagonists targeting the type II IFN-induced downstream events by generating a luciferase-based reporter system using IRF1[40] and NLRC5 promoters ("Methods"). Among the SARS-CoV-2 genes we tested, ORF6 showed the most dramatic inhibitory effect (87% reduction) on IFNγ-induced IRF1 GAS promoter activity (Fig. 4a). Similarly, we observed up to 83% reduction in IFNγ-induced NLRC5 promoter activity by overexpression of ORF6 (Fig. 4b). Nsp1, ORF8, and N proteins exhibited marginal inhibitory effects on these promoter activities. Presumably, this could be the result of partial reduction of the nuclear STAT1 level by Nsp1, ORF8, and N under IFNγ stimulation conditions (Fig. 3d). Further analysis by qPCR corroborated that the IFNγ-induced NLRC5 gene expression was significantly suppressed by ORF6 (Fig. 4c). To evaluate whether ORF6 can affect IRF1 function as a transcriptional regulator of the MHC class I pathway, we examined IRF1-mediated MHC class I expression using a reporter assay system that featured luciferase reporters containing proximal HLA promoters[41,42]. We confirmed that overexpression of IRF1 could induce the activity of HLA-A, B, and C promoters. However, this induction was significantly suppressed by ORF6 (Fig. 4d). Therefore, these data indicate that SARS-CoV-2 ORF6 not only inhibits IFNγ-mediated gene expression of IRF1 and NLRC5 but also interferes with IRF1 function.

To explore the mechanism by which ORF6 inhibits the IRF1 function, we evaluated the nuclear translocation of IRF1 upon IFNγ stimulation. Compared to empty vector expressing control cells, ORF6 positive cells showed a dramatic reduction in nuclear IRF1 levels under type II IFN stimulated conditions (Fig. 4e). We also observed that ORF8 could marginally suppress IRF1 nuclear importation, and this could be a result of the slight suppression of STAT1 nuclear import by ORF8 (Fig. 3d). Altogether, these data suggest that SARS-CoV-2 ORF6 is a primary viral antagonist targeting type II IFN-mediated immune responses.

**SARS-CoV-2 ORF6 directly inhibits CITA function of NLRC5.** Our analysis of SARS-CoV-2 inhibition on the MHC class I

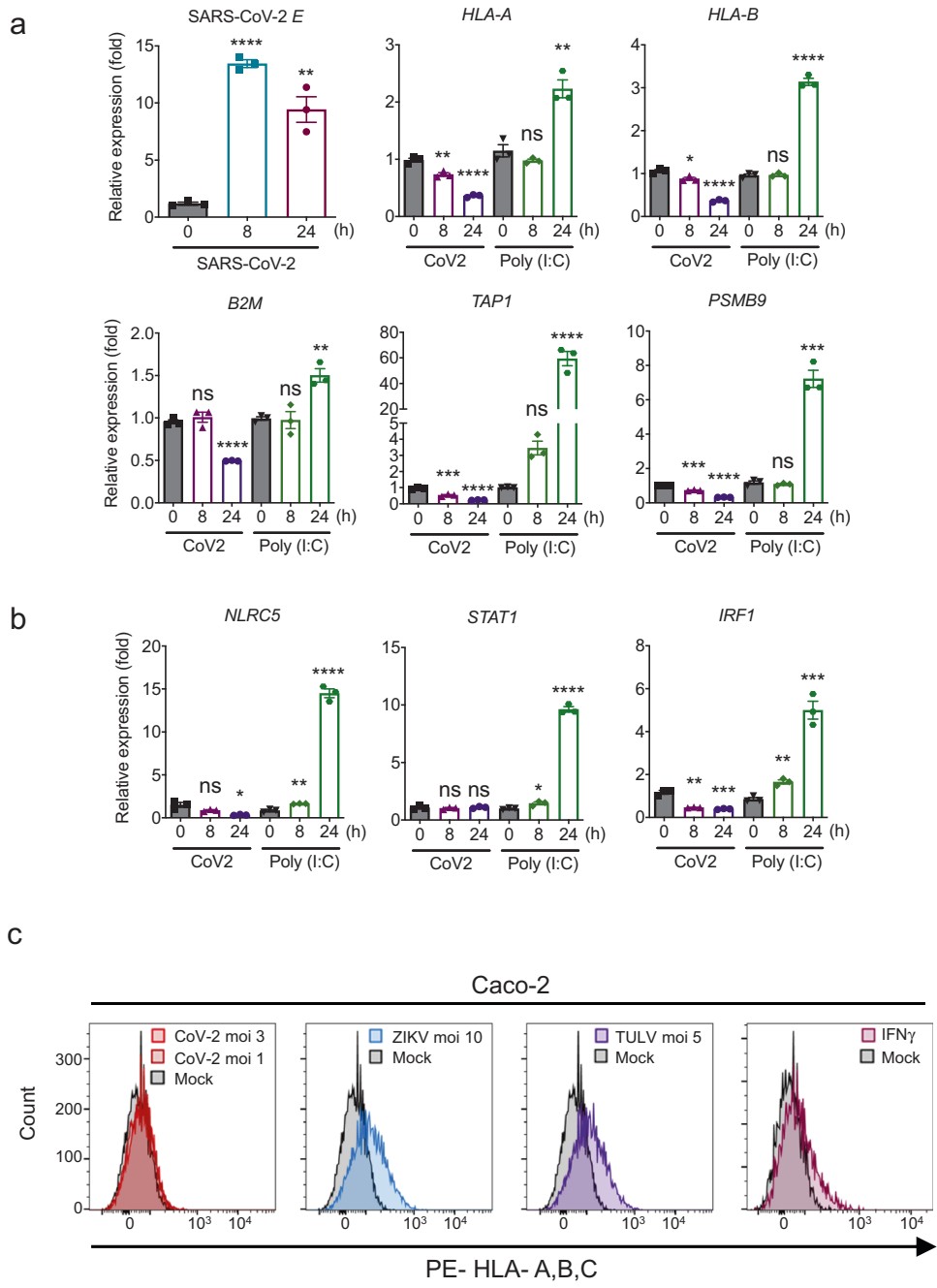

**Fig. 2 SARS-CoV-2 inhibits upregulation of MHC class I pathway genes in epithelial cells. a** Quantitative real-time PCR analysis of the transcript levels of the MHC class I components upon SARS-CoV-2 infection (moi 3) or poly(I:C) transfection (200 ng) in Calu-3 cells. The results are from three independent experiments. **b** Gene expression of the transactivator and transcription factors of the MHC class I pathway upon SARS-CoV-2 infection (moi 3) or poly(I:C) transfection (200 ng) in Calu-3 cells analyzed by quantitative real-time PCR. The results are from three independent experiments. **c** Surface expression of MHC class I upon SARS-CoV-2, ZIKV, and TULV infection with indicated moi for 48 h in Caco-2 cells. Stimulation with IFNγ (100 U/ml) for 48 h was used as a positive control. The FACS gating strategies are provided in Supplementary Fig. 4.

pathway showed not only inhibition of upregulation of expression but also reduction of the expression of MHC class I-related genes by SARS-CoV-2 infection (Figs. 1 and 2). Therefore, we reasoned that in addition to the impaired upregulation of NLRC5, there might be other mechanisms that directly block the transcriptional regulation of the MHC class I genes. Given that NLRC5 is a key CITA, we asked whether SARS-CoV-2 protein can directly inhibit NLRC5 CITA function. To screen the specific SARS-CoV-2 proteins that suppress NLRC5 activity, we utilized the HLA-B250 luciferase reporter system containing the proximal HLA-B

promoter in HEK293T cells. First, we confirmed that over-expression of the wild type (WT) NLRC5, but not a functionally defective mutant (K234A)[13], could induce the HLA-B250 promoter activity up to 10-fold. Under this condition, we found that SARS-CoV-2 ORF6, but not other viral genes tested, remarkably suppressed the NLRC5-mediated HLA-B250 promoter activity (Fig. 5a). This inhibitory effect was also confirmed by the Calu-3 cell line (Supplementary Fig. 8). The RT-qPCR analysis further confirmed that the expression level of the MHC class I-related genes induced by NLRC5 was significantly suppressed by ORF6

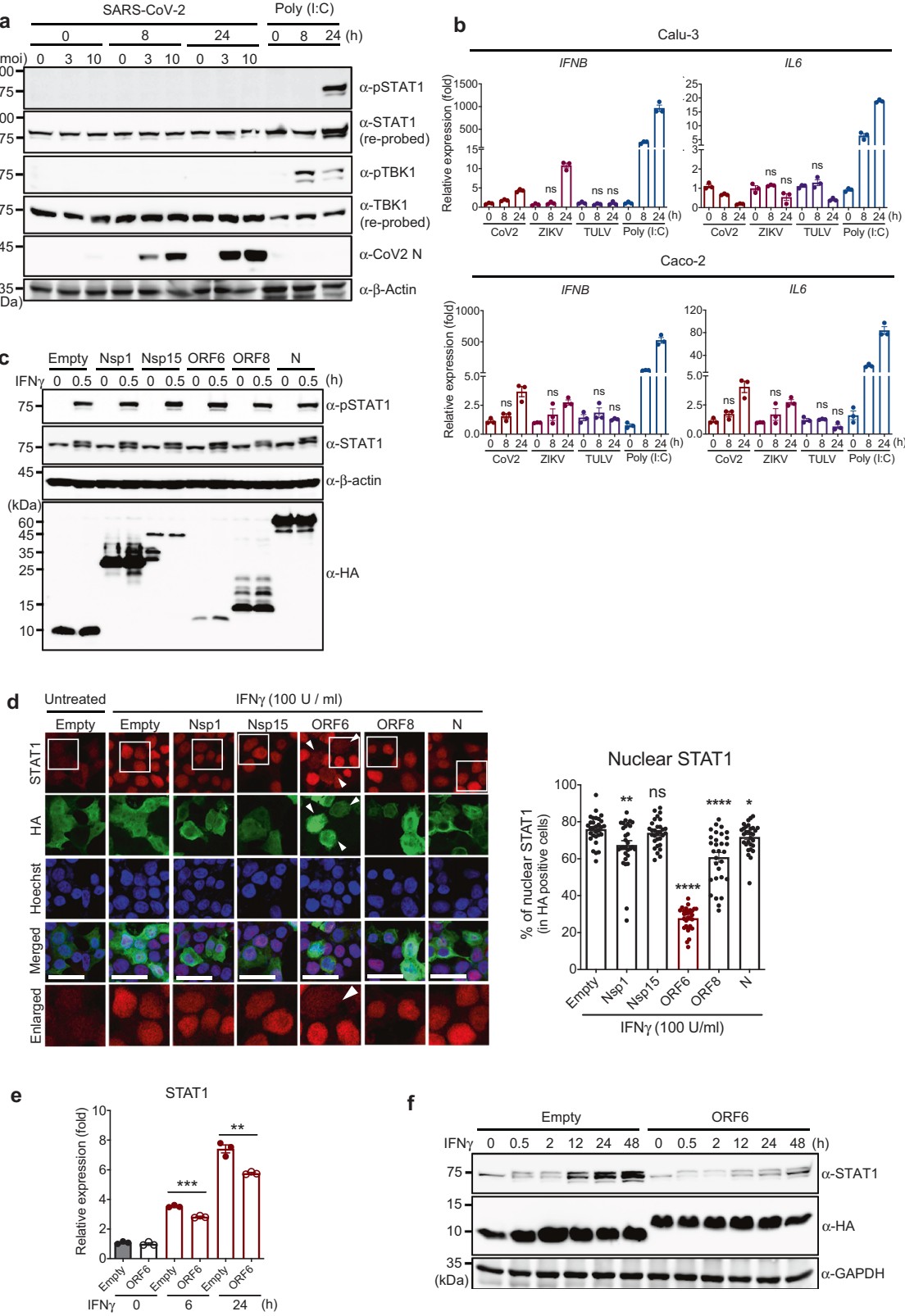

(Fig. 5b). Next, we monitored NLRC5-mediated induction of MHC class I surface expression by flow cytometry analysis. NLRC5-expressing cells showed an increased HLA-A, B, C surface expression. However, overexpression of ORF6 completely blocked NLRC5-mediated HLA-A, B, C surface expression (Fig. 5c and Supplementary Fig. 9). Recent data showed that SARS-CoV-2 ORF8 could inhibit the MHC class I pathway

through the autophagy-mediated degradation of the MHC class I-related molecules[43]. Therefore, we examined the inhibitory role of SARS-CoV-2 ORF8 in MHC class I induction. Although SARS-CoV-2 ORF8 could partially suppress the type II IFN signaling pathway (Figs. 3 and 4), we did not observe any inhibitory effect of SARS-CoV-2 ORF8 on the NLRC5-mediated MHC class I induction when evaluating both promoter activity and surface

**Fig. 3 SARS-CoV-2 ORF6 inhibits IFNγ-mediated STAT1 function. a** Immunoblot analysis of the indicated protein level in Calu-3 cells infected with SARS-CoV-2 at indicated moi or stimulated with poly(I:C) (200 ng) for the indicated time. The data shown is the representative result from two independent experiments. **b** Quantitative real-time PCR analysis of the indicated gene expression level in SARS-CoV-2 (moi 3), ZIKV (moi 10), and TULV (moi 5)-infected, or poly(I:C) (200 ng) transfected Calu-3 or Caco-2 cells for the indicated time. The results are from three independent experiments. **c** Immunoblot analysis of the indicated protein level in IFNγ (100 U/ml) treated HEK293T cells expressing the indicated SARS-CoV-2 proteins. The data shown is the representative result from two independent experiments. **d** Immunofluorescence analysis of endogenous STAT1 cellular localization in IFNγ (100 U/ml) treated HEK293T cells expressing the indicated SARS-CoV-2 proteins or empty control. Images were obtained using a confocal microscope. The scale bar indicates 50 microns. Suppressed STAT1 nuclear localization is indicated with white arrows. White boxes indicate the area shown in Enlarged row. Quantitative comparison of the nuclear STAT1 signal intensity (% of nuclear signal intensity/total cell signal intensity) is shown with a bar graph analyzed by ImageJ. **e** Quantitative real-time PCR analysis of STAT1 gene expression level in SARS-CoV-2 ORF6 expressing HEK293T cells stimulated with IFNγ (100 U/ml) for the indicated time. The results are from three independent experiments. **f** Immunoblot analysis of the indicated protein level in SARS-CoV-2 ORF6 expressing HEK293T cells stimulated with IFNγ (100 U/ml) for the indicated time. The data shown is the representative result from two independent experiments.

expression (Fig. 5a and c). Moreover, we did not observe that ORF8 has an inhibitory effect on MHC class I expression at a steady state (Supplementary Fig. 10). While it is not clear why the inhibitory effect of ORF8 on MHC class I expression was observed previously, it might be possible that the inhibitory effect of ORF8 could only be observed under specific conditions.

**SARS-CoV-2 ORF6 inhibits NLRC5 nuclear transportation**. Previously, we showed that nuclear localization of NLRC5 is critical for its function as CITA[13]. Therefore, we asked whether inhibition of NLRC5 function by SARS-CoV-2 ORF6 is through blocking of NLRC5 nuclear translocation. To monitor the cellular localization of NLRC5, we generated a FLAG-NLRC5 stably expressing HeLa cell line (FLAG-NLRC5 HeLa). Since NLRC5 can shuttle between the cytoplasm and the nucleus, cells were treated with Leptomycin B, a drug that prevents the protein exportation from the nucleus to the cytoplasm, to secure the nuclear-localized NLRC5[13]. While other SARS-CoV-2 proteins such as Nsp1 and ORF8 did not affect NLRC5 nuclear localization, ORF6 expressing cells showed a significantly reduced nuclear NLRC5 level (Fig. 5d). Consistent results were also obtained using various cell lines such as A549, Calu-3, Caco-2, and HEK293T (Supplementary Fig. 11). By subcellular protein fractionation analysis using FLAG-NLRC5 HeLa cells, we also observed that the amount of nuclear NLRC5 was reduced by more than 50% in ORF6 overexpressed cells compared to control confirmed by Western blotting analysis (Fig. 5e). Notably, the electrophoretic mobility of NLRC5 protein from the cytoplasmic fraction was slightly slower than its mobility from the nuclear fraction (Fig. 5e, see asterisks). Presumably, NLRC5 might undergo posttranslational modification that may be required for promoting its nuclear translocation or function in the nucleus. However, the mechanistic nature of this regulation remains to be elucidated. Collectively, these data indicate that SARS-CoV-2 ORF6 prevents NLRC5 nuclear importation and thereby inhibits NLRC5 function.

**NLRC5 utilizes a karyopherin complex for nuclear import**. Our previous study showed that NLRC5 possesses a bipartite nuclear localization signal (NLS) motif[13]. NLRC5 with mutations in the NLS motif by replacing the residues, arginine$_{132}$, arginine$_{133}$, and lysine$_{134}$ (NLS I) with alanine (RRK$_{132/133/134}$A) or lysine$_{121}$ and arginine$_{122}$ (NLS II) with alanine (RK$_{121/122}$A) resulted in complete loss of NLRC5 nuclear localization (Fig. 6a). It has been shown that NLS-containing proteins are recognized by karyopherin alpha subunits (KPNAs) to form a trimeric 'cargo' complex with a karyopherin beta subunit 1 (KPNB1). This complex is then transported into the nucleus through the nuclear pore complex (NPC). Transcription factors, such as STAT1 and IRF1, were shown to interact with KPNA1 and KPNA2, respectively, for nuclear import[44,45]. Because NLRC5 has NLSs, we

reasoned that the nuclear import of NLRC5 might be regulated by the karyopherin-cargo complex. Thus, to test whether NLRC5 can interact with KPNAs, we performed immunoprecipitation analysis using various KPNA subunits from KPNA1 to KPNA6. We found that NLRC5 could interact with KPNA1 and KPNA6 (Fig. 6b). Notably, two distinct bands were detected from the immunoprecipitated KPNA1 and KPNA6. The same phenomena have been reported by previous studies with an unknown molecular mechanism[46,47]. To investigate whether the karyopherin complex is required for NLRC5 nuclear transport, we performed immunofluorescence analysis using RNA interference (RNAi) or ectopic expression of a dominant-negative mutant system. Depletion of KPNB1 and KPNA6 by siRNA resulted in impaired NLRC5 nuclear import compared to control siRNA treated-cells (Fig. 6c). Moreover, while the nuclear localization of NLRC5 was intact with WT KPNB1 expression, a dominant-negative form of KPNB1[48] abolished nuclear import of NLRC5 (Fig. 6d), corroborating that NLRC5 utilizes a karyopherin complex for its nuclear import.

**SARS-CoV-2 ORF6 inhibits nuclear import of KPNAs**. It has been shown that ORF6 of SARS-CoV and SARS-CoV-2 target the karyopherin-mediated nuclear import pathway and suppress the host type I IFN system[32,33,44]. Our data showed that SARS-CoV-2 ORF6 inhibits nuclear importation of NLRC5, which is mediated by karyopherin proteins. Therefore, we asked if the inhibitory effect of ORF6 on NLRC5 nuclear import was due to the dissociation of the NLRC5-karyopherin complex by ORF6. To address this question, we performed an immunoprecipitation assay to monitor the effect of ORF6 on the NLRC5 import complex. As shown in Fig. 6B, we confirmed the binding of NLRC5 to KPNA1/6 (Fig. 6e, HA group). However, we found that the expression of ORF6 did not interfere with this interaction (Fig. 6e, HA-ORF6 group). Notably, we found that ORF6 did not bind to NLRC5 (Fig. 6e, HA blotting). Thus, these results implicate that rather than directly inhibiting the NLRC5-karyopherin complex, ORF6 might target a downstream event in nuclear translocation. This notion agrees with a recent study showing that SARS-CoV-2 ORF6 was directly associated with one of the NPC components, Nup98, resulting in reduced interaction between Nup98 and KPNB1[49]. Indeed, we found that ORF6 could block the nuclear transport of multiple KPNA subunits (Fig. 6f). Therefore, even without direct interaction with NLS-containing cargo proteins such as NLRC5, SARS-CoV-2 ORF6 may exert its inhibitory role in nuclear import of the target cargo proteins by blocking karyopherin complex transport.

**The carboxy-terminus of ORF6 is required for inhibition of NLRC5**. A recent study reported that the C-terminus acidic

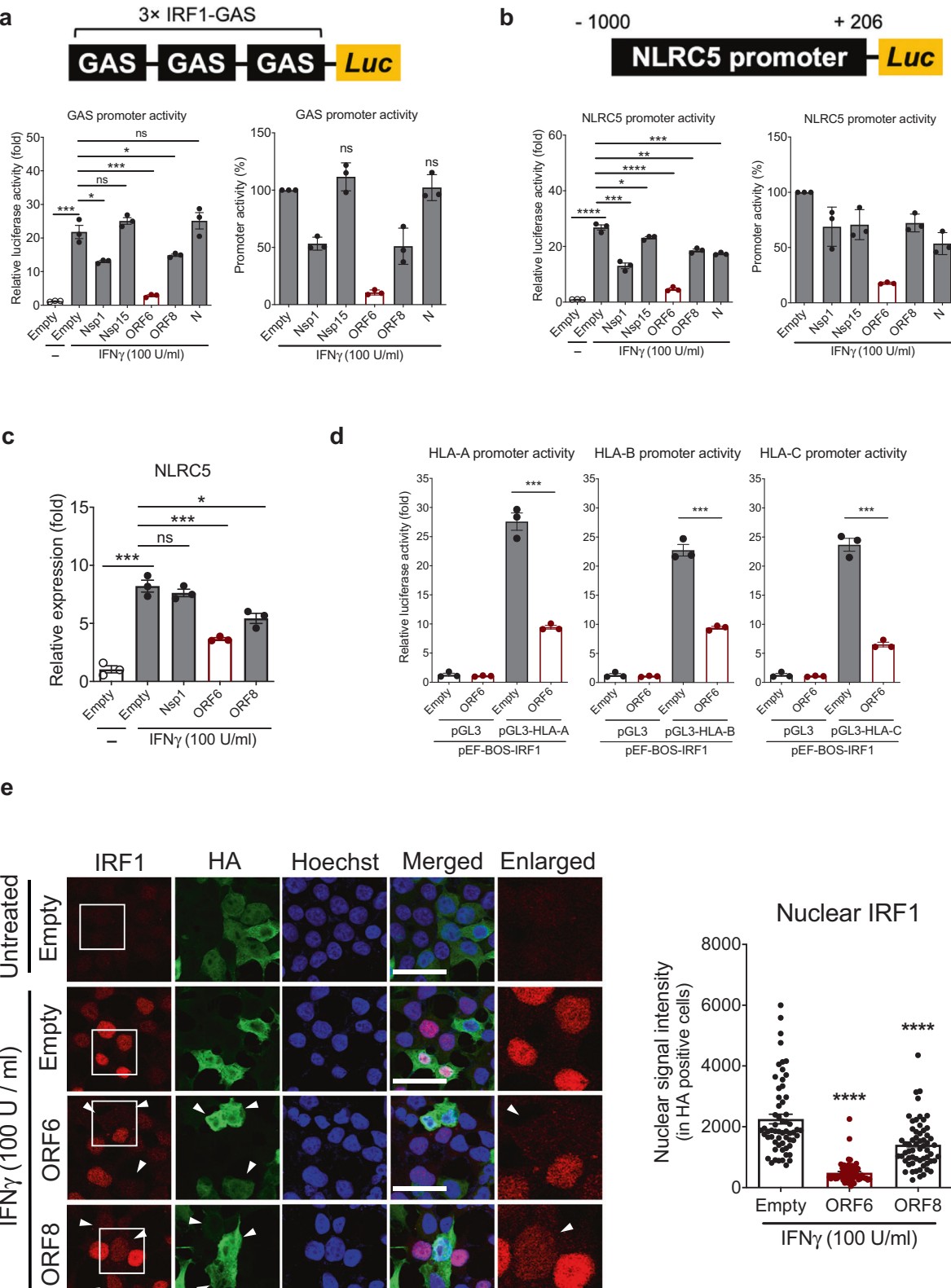

domain of SARS-CoV-2 ORF6 is critical for its inhibitory effect on STAT1 function[32]. Therefore, we analyzed the role of the C-terminus of ORF6 in the inhibition of NLRC5 function. We generated an ORF6 mutant in which the last six amino acids, glutamine$_{56}$, proline$_{57}$, methionine$_{58}$, glutamic acid$_{59}$, iso-leucine$_{60}$, and aspartic acid$_{61}$, were replaced with alanine (ORF6 6A) and compared its activity to ORF6 WT. We observed that

ORF6 WT, but not ORF6 6A, could strongly suppress NLRC5-mediated HLA-B250 promoter activity (Fig. 7a). The surface expression of HLA-A, B, C was remarkably suppressed by ORF6 WT, whereas ORF6 6A failed to show an inhibitory effect (Fig. 7b and Supplementary Fig. 12). Analysis of NLRC5 cellular locali-zation by immunofluorescence showed that while ORF6 WT could prevent NLRC5 nuclear transport, ORF6 6A no longer

**Fig. 4 SARS-CoV-2 ORF6 inhibits expression of IRF1 and NLRC5. a** and **b** Effect of SARS-CoV-2 proteins on the IFNγ-mediated IRF1 GAS or NLRC5 promoter activity. Schematic images illustrate the reporter assay systems for monitoring the promoter activity of IRF1 GAS and NLRC5. HEK293T cells were transfected with IRF1 3 × GAS- (**a**) or NLRC5 promoter-containing reporter construct (**b**), along with an empty control plasmid or with plasmids expressing the indicated SARS-CoV-2 proteins. At 24 h after transfection, cells were treated with IFNγ for 9 h, and luciferase activity was measured. The results are from three independent experiments. **c** Quantitative real-time PCR analysis of NLRC5 gene expression level in IFNγ (100 U/ml, 9 h) treated HEK293T cells expressing the indicated SARS-CoV-2 proteins. The results are from three independent experiments. **d** Effect of SARS-CoV-2 ORF6 on the IRF1-mediated HLA promoter activity. HEK293T cells were transfected with the indicated HLA reporter constructs, along with a IRF1 expression vector and the plasmid expressing empty or SARS-CoV-2 ORF6. At 36 h after transfection, cells were collected to measure the luciferase activity. The results are from three independent experiments. **e** Immunofluorescence analysis of the endogenous IRF1 cellular localization in IFNγ (100 U/ml, 6 h) treated HEK293T cells expressing the indicated SARS-CoV-2 proteins. Images were obtained using a confocal microscope. The scale bar indicates 50 microns. Suppressed expression and nuclear import of IRF1 is indicated with white arrows. A quantitative comparison of the nuclear IRF1 signal intensity (%) is shown with a bar graph using ImageJ.

retained its inhibitory activity (Fig. 7c). Furthermore, ORF6 WT, but not ORF6 6A, could suppress nuclear import of KPNB1, thereby suppressing NLRC5 nuclear transport (Fig. 7d). Altogether, these data suggest that the C-terminus region of SARS-CoV-2 ORF6 is critical to exert its antagonistic activity on the NLRC5 function.

## Discussion

In this study, we showed that the induction of MHC class I gene expression is impaired by SARS-CoV-2 infection. Gene expression studies in the lung and airway epithelial cells of COVID-19 patients as well as SARS-CoV-2 infected epithelial cell lines showed that SARS-CoV-2 inhibits the upregulation of MHC class I in epithelial cells during virus infection. Since MHC class I is critical for antiviral immunity, the MHC class I pathway has been known as a favorable target for immune evasion by various viral strains. Our data indicate that impaired upregulation of the MHC class I gene expression in the airway and intestinal epithelial cells during SARS-CoV-2 infection interferes with the CD8 T cell-dependent cellular immunity, thus causing a higher risk of exacerbation of viral loads and prolonged infection[4–6,50,51].

Intriguingly, SARS-CoV-2 targets NLRC5 at both transcriptional and functional levels to escape from the MHC class I pathway. Since NLRC5 orchestrates the concerted expression of the major components involved in the MHC class I pathway, it has been postulated that NLRC5 would be an attractive target for immune evasion[52–54]. Considering that inhibition of NLRC5 alone can lead to suppression of the MHC class I pathway, it is reasonable to speculate that targeting NLRC5 would be much more efficient than suppressing each component of the MHC class I pathway. In fact, in the case of cancer, in which MHC class I also plays crucial roles, NLRC5 was found to be the most critical host factor targeted by cancer cells to evade anti-cancer immunity. For example, epigenetic or genetic alterations such as promoter methylation, genetic mutations, or copy number loss of NLRC5 gene can cause impaired expression or function of NLRC5, associated with reduced MHC class I expression, impaired CD8 cell recruitment, and poor survival of cancer patients[52]. In this regard, it is reasonable to hypothesize that the expression level of NLRC5 in COVID-19 patients might be associated with the disease severity and mortality. It would be interesting to see if the genetic variability in the expression of NLRC5 and other MHC class I-related molecules are associated with the pathogenesis of SARS-CoV-2 and other viral infectious agents. Further study is required to address this speculation.

NLRC5-dependent MHC class I gene expression is induced by both type I, and more potently type II interferon[10,11]. It appears that SARS-CoV-2 can inhibit both signaling pathways. As shown previously and also confirmed in this study, the type I interferon signaling pathway is inhibited by SARS-CoV-2. Besides, STAT1 activation can be inhibited even in the presence of IFNγ, the most

potent inducer of STAT1 activation, indicating that SARS-CoV-2 strongly suppresses the type II IFN signaling pathway. As NLRC5 induction is STAT1-dependent, impaired STAT1 function directly leads to transcriptional suppression of the MHC class I genes. In addition to NLRC5, the induction of IRF1, another co-transcription factor involved in the MHC class I activation, is suppressed because IRF1 expression is also STAT1-dependent.

In addition to transcriptional suppression of NLRC5 expression, we found direct inhibition of NLRC5 function by a SARS-CoV-2 protein (Fig. 5). Screening of inhibitory molecules of SARS-CoV-2 showed that SARS-CoV-2 ORF6 inhibits NLRC5 CITA function. Cellular localization studies indicate that this is likely to be achieved by the inhibition of nuclear translocation of NLRC5. Likewise, we found that SARS-CoV-2 ORF6 also inhibits IRF1 function by blocking the IRF1 nuclear import. Inhibition of nuclear translocation of NLRC5 and IRF1 is reminiscent of the recent findings of the inhibitory nature of ORF6 of SARS-CoV and SARS-CoV-2 in innate immunity.

Recently it was shown that ORF6 could block the nuclear translocation of several transcription factors such as STAT1 and IRF3 by inhibiting karyopherin-mediated nuclear import, thereby suppressing immune responses[32,33].

Although all three transcription factors, STAT1, IRF1, and NLRC5, commonly utilize the karyopherin complex for nuclear translocation, their partner karyopherin alpha subunits are different. Nevertheless, SARS-CoV-2 ORF6 can simultaneously block the nuclear import of all three factors, suggesting that SARS-CoV-2 ORF6 may exert its inhibitory role by targeting the downstream events of the karyopherin-mediated nuclear import. In fact, during our preparation for the manuscript, one study has demonstrated that SARS-CoV-2 ORF6 directly binds to Nup98, a component of the nuclear pore complex, and prevents the karyopherin-cargo complex-mediated nuclear transportation[49]. In agreement with this study, we found that SARS-CoV-2 ORF6 suppressed the nuclear localization of all karyopherin alpha subunits that are responsible for nuclear import of STAT1, IRF1, and NLRC5 regardless of upstream molecule interaction (Fig. 6f).

In this regard, it should be noted that although SARS-CoV ORF6 and SARS-CoV-2 ORF6 have a similar inhibitory effect, the mechanistic nature of their inhibitory function seems to be different. As mentioned above, whereas SARS-CoV-2 ORF6 blocks the nuclear import of multiple KPNA subunits (Fig. 6f), SARS-CoV ORF6 only selectively inhibits the nuclear import of KPNA2 via direct interaction[44]. It was shown that while SARS-CoV-2 ORF6 strongly interacts with Nup98[49], SARS-CoV ORF6 showed a marginal association[55]. Instead, SARS-CoV ORF6-KPNA2 complex competes with other KPNA subunits to hijack KPNB1 from the target cargo molecule[44]. This mechanistic difference between SARS-CoV ORF6 and SARS-CoV-2 ORF6 could explain why MHC class I activation can be efficiently suppressed by SARS-CoV-2 but not by SARS-CoV[23]. We and others showed

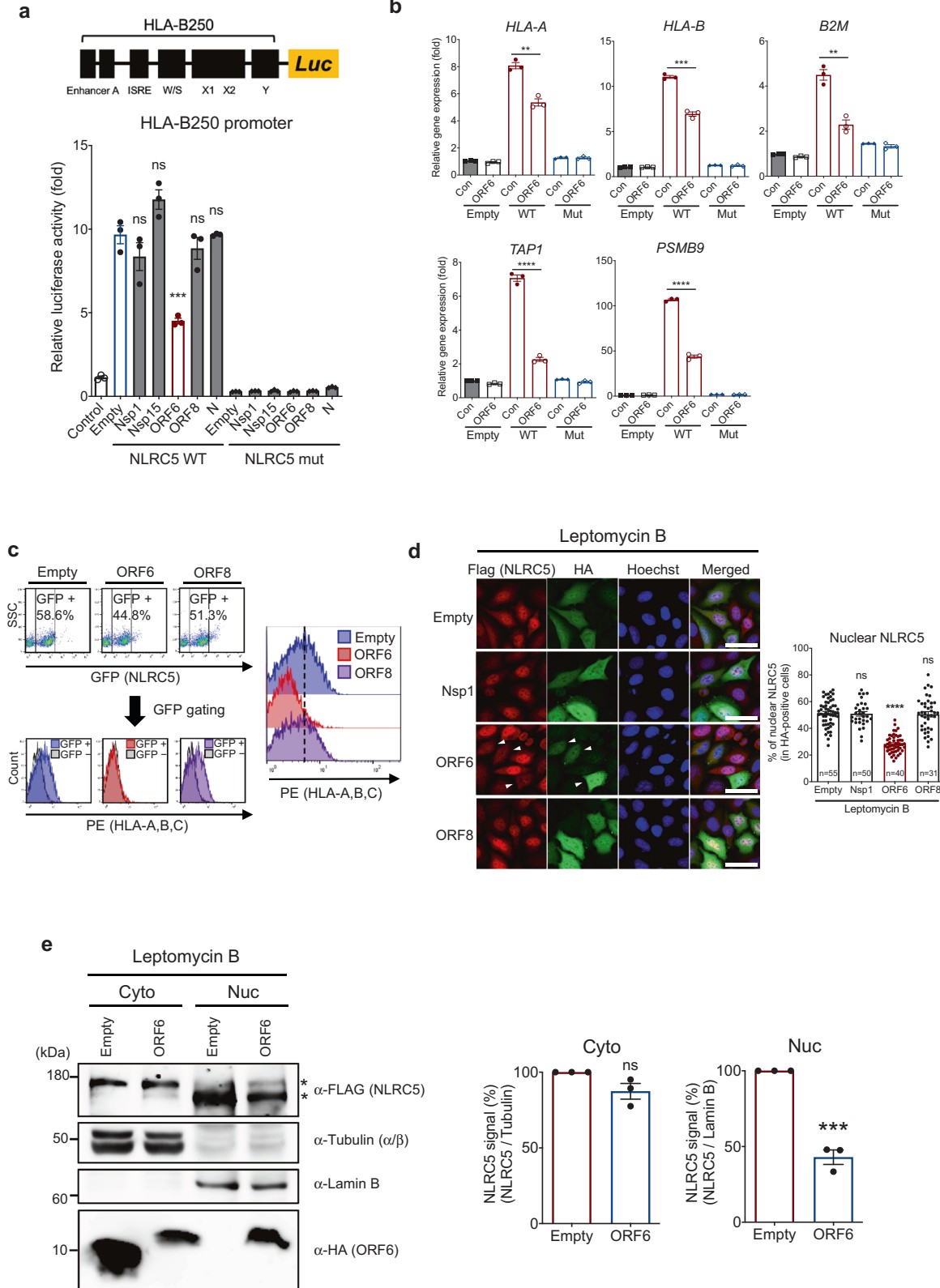

that the C' terminus region of ORF6 is critical for its inhibitory function (Fig. 7a–d). Notably, the amino acid sequence of C' terminus regions between SARS-CoV ORF6 and SARS-CoV-2 ORF6 is slightly different. Whether or not the mechanistic differences between these two ORF6s are due to the difference in amino acid sequence at their C' terminus tail still needs to be clarified.

It should be noted that while ORF6 displays a strong inhibitory effect on the upregulation of MHC class I under the stimulated conditions such as IFNγ treatment, it does not seem to have much impact on the expression of the MHC class I under the steady state condition (Supplementary Fig. 10). Thus, the inhibitory effect of ORF6 alone cannot explain how the MHC class I gene expression was downregulated in the SARS-CoV-2-infected

**Fig. 5 SARS-CoV-2 ORF6 suppresses NLRC5 CITA function. a** Effect of SARS-CoV-2 proteins on NLRC5-mediated HLA-B promoter activity. The schematic image illustrates the reporter assay system for monitoring the promoter activity of HLA-B. HEK293T cells were transfected with HLA-B proximal promoter reporter construct and plasmids expressing WT or Walker A mutant NLRC5, along with a control plasmid, or with plasmids expressing the indicated SARS-CoV-2 proteins. At 36 h after transfection, cells were collected to measure the luciferase activity. The results are from three independent experiments. **b** Effect of SARS-CoV-2 ORF6 on NLRC5-mediated expression of the MHC class I-related genes in HEK293T cells analyzed by quantitative real-time PCR. The results are from three independent experiments. **c** Effect of SARS-CoV-2 proteins on NLRC5-mediated surface expression of HLA- A, B, and C in HEK293T cells analyzed by flow cytometry. GFP (NLRC5) positive cells were gated for evaluating PE (HLAs) signal intensity. The FACS gating strategies are provided in Supplementary Fig. 9. **d** Effect of SARS-CoV-2 proteins on the cellular localization of FLAG-tagged NLRC5 in leptomycin B (100 nM, 8 h) treated HeLa cells stably expressing FLAG-NLRC5 by immunofluorescence analysis. Representative images show reduced NLRC5 nuclear localization indicated with white arrows. Quantitative comparison of the nuclear NLRC5 signal intensity (% of nuclear signal intensity/total cell signal intensity) is shown with a bar graph analyzed by ImageJ. Images were obtained using a confocal microscope. The scale bar indicates 50 microns. The sample numbers for evaluation are indicated in the bar graph. **e** Immunoblot analysis of the indicated proteins from the cell lysates of the nuclear or cytosolic fraction in either control or SARS-CoV-2 ORF6 expressing FLAG-NLRC5 HeLa stable cells. Asterisks indicate two distinct bands of NLRC5. Quantitative comparison of the cytosolic or nuclear FLAG-NLRC5 signal intensity normalized by the intensity of Lamin B1 (for nuclear NLRC5) or α/β-tubulin (for cytosolic NLRC5) is shown with a bar graph via analysis with ImageJ. The data shown is a representative result from three independent experiments.

patient samples (Fig. 1a–c). In this regard, multiple mechanism, presumably, including other SARS-CoV-2 proteins, may be involved in the downregulation of the MHC class I expression during SARS-CoV-2 infection. Further study is required to address the more detailed function of ORF6 and other possible inhibitory mechanisms.

Does SARS-CoV-2 ORF6 inhibit the nuclear import of all the nuclear-localized proteins? Since NLS-containing cargo proteins utilize karyopherin complex-mediated regulation for their nuclear translocation, SARS-CoV-2 ORF6 may broadly, but specifically, block the function of the NLS-containing proteins. Yet, there is a karyopherin-independent regulation of nuclear import via ankyrin domain-mediated mechanism. Interestingly, a recent study identified a molecular code termed RaDAR (RanGDP/AR) that regulates ankyrin repeat-mediated nuclear transportation[56]. Importantly, the authors further discovered that transcriptional regulators involved in pro-inflammatory responses, such as NF-κB or p53 pathways, generally utilize the RaDAR regulation pathway for nuclear import. Considering that whereas COVID-19 patients failed to produce type I and III IFNs, still pro-inflammatory cytokines are robustly generated. It would be interesting if this is due to the fact that SARS-CoV-2 does not affect the nuclear transport of those ankyrin repeat-containing transcriptional regulators. Further study is required to address this interesting question.

Altogether, our study provides mechanistic insight into ORF6-mediated SARS-CoV-2 immune evasion via targeting the MHC class I antigen-presenting pathway (Fig. 7e). Moreover, our study indicates that SARS-CoV-2 possesses impressive immune evasion strategies targeting two critical host antiviral defense programs, the MHC class I and IFN signaling pathways, that result in successful viral adaptation to human hosts. The mechanistic findings in this study may provide potential molecular targets for developing the therapeutics against COVID-19.

## Methods

**Data mining and bioinformatic analyses**. Raw counts and metadata were accessed and downloaded from the NCBI Gene Expression Omnibus (GEO) Repository, GSE152075 (patient nasopharyngeal swab data)[24] and GSE147507 (normal human bronchial epithelial cell [NHBE] data)[26], respectively. Analysis was performed using R 3.6.0 and RStudio 1.1.463. dplyr[57] 0.8.4, plyr[58] 1.8.4, and stringr[59] 1.4.0 packages were used for data manipulation. Raw count data were normalized using DESeq2[60] 1.26 with default settings and filtered to keep only genes with more than a total of ten counts across all samples. For the NP swab dataset, SARS-CoV-2 infection status was used as the design formula to obtain normalized *PTPRC (CD45)* gene expression values. To minimize the potential influence of SARS-CoV-2 infected patients with high immune cell influx (high *PTPRC* expression) on MHC class I gene expression, we used the expression range of *PTPRC* in non-infected patients as inclusion criteria (lower bound = 0, upper

bound = 75% quartile + IQR*1.5). For the NHBE dataset, the experimental condition was used as the design formula. Heatmap of z-score transformed log2 normalized count gene expression was generated using the ComplexHeatmap 2.2.0 package[61]. Violin plots, boxplots, bar plots, scatter plots, and linear regression lines were generated using ggplot2[62] 3.3.2, ggbeeswarm[63] 0.6.0, and ggpubr[64] 0.2.5. Fold changes and Benjamini–Hochberg adjusted two-sided Wald-test *P*-values depicted in Fig. 1c and d for MHC class I genes were extracted from differential gene analysis tests using DESeq2. For Supplementary Fig. 1b, 10X single-cell RNA-seq data from normal human lung tissue published by Travaglini et al.[25] was accessed and analyzed using the cellxgene platform (https://cellxgene.cziscience.com/e/krasnow_lab_human_lung_cell_atlas_10x-1-remixed.cxg/). Data was subset to include only cells from the lung tissue. The uniform manifold approximation projection (UMAP) visualization was annotated with metadata for broad cell types using the "compartments" data field, and expression data for individual genes (*PTPRC, NLRC5, HLA-A, HLA-B, HLA-C*) was interrogated. For Supplementary Fig. 2, log2 fold change data was accessed from the provided S2 Data published by Ravindra et al.[27].

**Cell lines**. Calu-3, HeLa, and HEK293T were from ATCC. A549, Caco-2, and Huh7 cells were from RIKEN Bio BRC, Japan. Cells were maintained at 37 °C in a humidified incubator with 5% $CO_2$. Calu-3 cells were maintained in Minimum Essential Medium Eagle (MEM) supplemented with 10% fetal bovine serum (FBS), penicillin–streptomycin (100 U/ml and 100 μg/ml, respectively), and 1% non-essential amino acids (NEAA). Caco-2 cells were maintained in RPMI 1640 supplemented with 10% fetal bovine serum (FBS), penicillin–streptomycin (100 U/ml and 100 μg/ml, respectively). A549, Huh7, HeLa, and HEK293T cells were maintained in Dulbecco's Modified Eagle Medium (DMEM) supplemented with 10% fetal bovine serum (FBS) and penicillin–streptomycin (100 U/ml and 100 μg/ml, respectively). FLAG-tagged human NLRC5-expressing stable cell line was generated with HeLa cells by using pQCXIP-FLAG-NLRC5 vector, followed by puromycin (3 μg/mL) selection. For human ACE2-expressing HEK293T stable cell line was generated by lentivirus system using pLVSIN-CMV-ACE2-Pur and selected with puromycin (3 μg/mL)[65].

**Virus infection and RNA transfection**. SARS-CoV-2 WK-521 strain was kindly provided by Dr. Shimojima (National Institute of Infectious Diseases, Japan)[66]. The original stock of this virus (wild type, WT) was prepared by inoculation of Vero-TMPRSS2 cells with Mynox mycoplasma elimination reagent (Minerva Biolabs). ZIKV (MR-766 strain) is provided by Dr. Takasaki (National Institute of Infectious Diseases, Japan). Tula virus (P-3245 strain)[67] is a kind gift of Dr. Yoshimatsu (Hokkaido University, Japan). For virus infection, cells were washed with PBS and treated with the culture medium (mock-treated) or infected with SARS-CoV-2 in the medium containing 2% FBS. After adsorption at 37 °C for 1 h, the medium was changed, and infection was continued for various periods with indicated virus titers in the presence of the serum-containing DMEM. Poly I:C was purchased from Amersham Biosciences (Arlington Heights). 5′-ppp RNA (5′- pppGGGAAACUGAAAGGGGAGAAGUGAAA-GUG-3′)[28] was synthesized by in vitro transcription using the AmpliScribe T7-Flash Transcription kit (Epicenter). RNAs were delivered into the cells with Lipofectamine 2000 according to the manufacturer's instructions (Invitrogen).

**Plasmids**. pcGN plasmid containing HA-tag (pcGN-HA) is a kind gift of Dr. Hatakeyama (Hokkaido University). SARS-CoV-2 genes used in this study were PCR amplified from either cDNA that generated using RNA from SARS-CoV-2 infected Caco-2 cells (for ORF8), or the construct templates purchased from Addgene [Nsp1 (Catalog #141367), Nsp15 (Catalog #141381), ORF6 (Catalog #141387), and N (Catalog #141391)] and provided by Dr. Figueiredo (Texas A&M

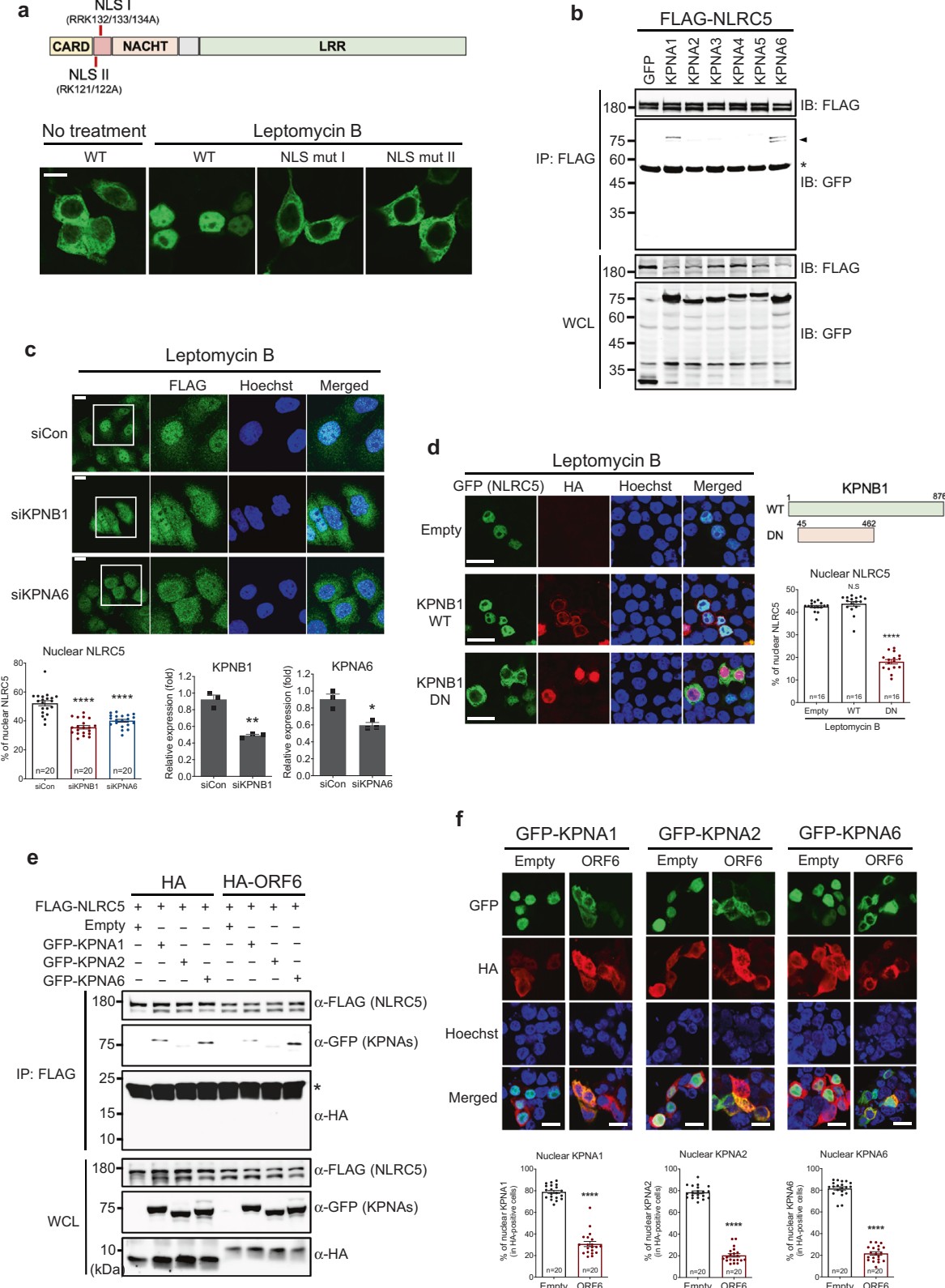

University, USA). Amplified viral genes were cloned into pcGN-HA using *BamH*I and *Xba*I. The mutant of ORF6 (ORF6 6 A) was generated by standard site-directed mutagenesis method by PCR using designed primer sets. IRF1 GAS reporter was generated by inserting IRF1 GAS elements repeated for three times as reported[40]. The GAS element DNA was synthesized and cloned into a pGL3 vector using *Kpn*I and *Xho*I sites. The reporter system of the NLRC5 promoter was generated by PCR amplification of the promoter region from −1000 to +206 flanked with the first exon of NLRC5 using genomic DNA from HeLa cells. The amplified DNA fragment was cloned into a pGL3 vector using *Kpn*I and *Xho*I sites.

The HLA promoters, pGL3-HLA-A, pGL3-HLA-B, and pGL3-HLA-C are kind gifts from Dr. van den Elsen (Leiden University, Netherlands)[42]. Human IRF1 expression vector, pEF-BOS-IRF1, is a kind gift from Dr. Fujita (Kyoto University, Japan). GFP-tagged NLRC5 WT, Walker A mutant, NLS mutant I, and NLS mutant II were previously reported[13]. Karyopherin alpha subunit expression vectors for KPNA2, KPNA3, KPNA4, and KPNA5 protein were generated by amplifying the coding sequences of KPNA2, KPNA3, KPNA4, and KPNA5 using cDNA generated using RNA from HeLa cells. The PCR amplified DNA fragments were cloned into the pcDNA3.1(+)-GFP vector (Invitrogen) using *BamH*I and

**Fig. 6 NLRC5 utilization of the karyopherin complex for nuclear import is targeted by SARS-CoV-2 ORF6. a** The cellular localization of WT or mutant NLRC5 by immunofluorescence analysis. The schematic image of the domain of NLRC5 protein illustrates the positions of two NLSs and the mutated sites at NLSs marked with the red bars. HEK293T cells were transfected with plasmids expressing GFP-tagged WT, NLS I mutant, or NLS II mutant NLRC5. At 24 h after transfection, cells were stimulated with 100 nM of leptomycin B for 8 h and then analyzed by confocal microscopy. The scale bar indicates 20 microns. The results shown are the representative data from the multiple images taken (2–4 images, $n = 2$ biological replicates) for each experiment. **b** Immunoblot analysis of the indicated proteins from the immunoprecipitation analysis using FLAG-tagged NLRC5 and GFP-tagged karyopherin alpha subunits. HEK293T cells were transfected with plasmids expressing FLAG-NLRC5, along with GFP alone or GFP-tagged KPNA subunits as indicated. After 48 h incubation, cell lysates were prepared and mixed with anti-FLAG M2 beads. Immunoprecipitated protein complexes were analyzed by SDS-PAGE. Karyopherin bands with smaller sizes are shown with an arrow. The asterisk indicates a heavy chain of the immunoglobulin used for IP. Two percent of the input proteins are shown in the whole cell lysate (WCL) panel. The data shown is the representative result from two independent experiments. **c** Immunofluorescence analysis of FLAG-tagged NLRC5 cellular localization in KPNB1 or KPNA6 depleted FLAG-NLRC5 stable HeLa cells by leptomycin B treatment (100 nM, 8 h). Quantitative comparison of the nuclear FLAG-NLRC5 signal intensity (% of nuclear signal intensity/whole-cell signal intensity) is shown with a bar graph analyzed by ImageJ (bottom left). The knockdown efficiency of the indicated genes targeted by the specific siRNAs was shown by comparing with gene expression level in the control siRNA transfected cells (bottom middle and right). The results are from three independent experiments. Images were obtained using a confocal microscope. The scale bar indicates 20 microns. The sample numbers for evaluation are indicated in the bar graph. **d** Immunofluorescence analysis of GFP-tagged NLRC5 cellular localization by WT or dominant-negative KPNB1. HEK293T cells were transfected with plasmids expressing GFP-tagged NLRC5, along with plasmids expressing HA-tagged WT, or dominant-negative (DN) KPNB1. At 24 h after transfection, cells were treated with 100 nM of leptomycin B for 8 h and then analyzed by confocal microscopy. The schematic image depicts the domain of WT or DN KPNB1 with amino acid numbers. Quantitative comparison of the nuclear NLRC5 signal intensity (% of nuclear signal intensity/total cell signal intensity) is shown with a bar graph analyzed by ImageJ. The sample numbers for evaluation are indicated in the bar graph. **e** Immunoblot analysis of the proteins from the immunoprecipitation analysis using FLAG-tagged NLRC5, HA-tagged empty or SARS-CoV-2 ORF6, and GFP-tagged karyopherin alpha subunits as indicated. HEK293T cells were transfected with plasmids expressing FLAG-NLRC5, along with HA alone or HA-tagged SARS-CoV-2 ORF6, and cotransfected with constructs expressing GFP-tagged KPNA subunits as indicated. After 48 h incubation, cell lysates were prepared and mixed with anti-FLAG M2 beads. Immunoprecipitated protein complexes were analyzed by SDS-PAGE. The asterisk indicates a light chain of the immunoglobulin used for IP. Two percent of the input proteins is shown in the WCL panel. The data shown is the representative result from two independent experiments. **f** Immunofluorescence analysis of the cellular localization of GFP-tagged karyopherin alpha subunits by HA-empty or HA-SARS-CoV-2 ORF6. HEK293T cells were transfected with the plasmids expressing GFP-tagged KPNA1, KPNA2, or KPNA6, along with plasmids expressing HA alone or HA-tagged SARS-CoV-2 ORF6. At 24 h after transfection, cells were analyzed by confocal microscopy. Quantitative comparison of the nuclear signal intensity (% of nuclear signal intensity/total cell signal intensity) of karyopherins is shown with a bar graph analyzed by ImageJ. The scale bar indicates 20 microns. The sample numbers for evaluation are indicated in the bar graph.

*Xba*I for KPNA2, *Bam*HI and *Xho*I for KPNA4, and *Eco*RI and *Xho*I for KPNA3 and KPNA5, respectively. KPNA1 and KPNA6 genes were amplified from the construct templates purchased from Addgene [pCMVTNT-T7-KPNA1 (Catalog #26677) and pCMVTNT-T7-KPNA6 (Catalog # 26682)] and cloned into the pcDNA3.1(+)-GFP vector using *Eco*RI and *Xba*I sites. The pEGFP-KPNB1 plasmid was purchased from Addgene (Catalog #106941). For HA-tagged WT and dominant-negative mutant KPNB1 (45-462)[48] are generated by PCR amplification using pEGFP-KPNB1 vector as a template, and amplified DNA fragments were cloned into the pcDNA3.1 vector using *Bam*HI and *Xba*I sites. The sequences of all of the cloned constructs were confirmed by Sanger sequencing. The sequence information of the primers for cloning is shown in Supplementary Table 1. Recombinant DNA experiments using SARS-CoV-2 genes were performed under confirmation of the Minister of Education, Culture, Sports, Science, and Technology (Approval number 2020-036, 2020-041) following the guidance of the Act on the Conservation and Sustainable Use of Biological Diversity through Regulations on the Use of Living Modified Organisms.

**Flow cytometry**. For analyzing surface expression level of HLA, $3 \times 10^5$ of human ACE2 stably expressing HEK293T or Caco-2 cells were infected with SARS-CoV-2 (multiplicity of infection (moi) 1 for ACE2-HEK293T and moi 1 or moi 3 for Caco-2), ZIKV (moi 10), and TULV (moi 5). As a control, cells were stimulated with 200 ng of the immunostimulatory RNAs (poly I:C or 5-'ppp RNA) by transfection or treated with IFNγ (100 U/ml) (Peprotech). For analyzing NLRC5-mediated HLA surface expression, GFP-tagged NLRC5 was transfected to $3 \times 10^5$ of HEK293T cells with HA-empty, HA-tagged SARS-CoV-2 ORF6, HA-tagged SARS-CoV-2 ORF8, or HA-tagged SARS-CoV-2 ORF6 6A mutant. After 48 h stimulation, cells were washed with PBS, and collected cells in FACS buffer (2% FBS/PBS) were stained with either PE-conjugated isotype control (BioLegend) or PE-conjugated anti-human HLA-A, B, C antibody (BioLegend). After 30 min incubation on ice, cells were washed twice with a FACS buffer and fixed with 4% paraformaldehyde (PFA) for 30 min at 4 °C. Finally, cells were washed twice with a FACS buffer and used for flow cytometric analysis with FACS Canto (BD). Data analysis was performed with FlowJo software (BD).

**Immunoprecipitation and western blotting**. For immunoprecipitation, $8.5 \times 10^6$ of HEK293T cells in 10 cm dish were harvested 48 h after polyethylenimine (PEI) transfection and lysed in the lysis buffer (50 mM Na-HEPES [pH 7.5], 150 mM NaCl, 1 mM EDTA, 0.5% Triton X-100, 10% glycerol, 1 mM PMSF, protease inhibitor cocktail [Roche], and phosphatase inhibitor [Nacalai Tesque]). Cell extracts were precleared with an empty protein A/G agarose (Pierce) for 1 h at 4 °C and subsequently incubated with anti-FLAG M2 affinity gel (Sigma) for 4 h at 4 °C. Immunoprecipitants were washed with Triton X-100 lysis buffer three times and resuspended in SDS sample buffer, boiled for 5 min at 95 °C, resolved on SDS/PAGE gels, and transferred onto polyvinylidene difluoride (PVDF) membranes. Antibody concentrations were as follows: anti-FLAG antibody (Sigma), 1:2000; anti-HA antibody (BioLegend), 1:2000; anti-TBK1 antibody (Cell Signaling), 1:1000; anti-phospho-TBK1 antibody (Cell Signaling), 1:1000; anti-STAT1 antibody (Cell Signaling), 1:1000; anti-phospho-STAT1 antibody (Cell Signaling), 1:1000; anti-SARS-CoV-2 N antibody (GeneTex), 1:1000; anti-GFP antibody (Proteintech), 1:5000; anti-Lamin B1 (Cell Signaling), 1:1000; anti-beta actin (Proteintech), 1:1000; anti-alpha Tubulin (Proteintech), 1:1000; anti-GAPDH (Proteintech), 1:1000; and secondary antibodies with horseradish peroxidase conjugate (GE Healthcare), 1:5000. Images were developed with ECL reagent (Takara) and scanned with ImageQuant™ LAS 4000 (GE Healthcare).

**Subcellular protein fractionation**. FLAG-NLRC5 stable HeLa cells were transfected with empty vector or HA-tagged SARS-CoV-2 ORF6 using Lipofectamine 2000. At 48 h post-transfection, the medium was aspirated, and cells were washed twice with phosphate-buffered saline (PBS). Cells were collected and scraped in cold fractionation buffer (1 mM EDTA, 1 mM EGTA, 10 mM KCl, 1.5 mM MgCl₂, 20 mM HEPES [pH 7.4], protease inhibitor cocktail [Roche], 1 mM dithiothreitol [DTT]), and incubate 15 min on ice. Cells were then homogenized by passing through a 26-gauge syringe ten times. After incubation on ice for 20 min, the homogenate was spun at $3000 \times g$ for 10 min, 4 °C, and the supernatant was transferred into a fresh tube. The nuclear pellet was washed with a fractionation buffer. The pellet was dispersed with a pipette and passed through a 25-gauge syringe ten more times, and the nuclear pellet was resuspended in nuclear buffer (standard lysis buffer with 10% glycerol and 0.1% SDS added). The supernatant was centrifuged at $10000 \times g$, and the supernatant was used as a cytosolic fraction. Both nuclear and cytosolic fractions were further analyzed by SDS-PAGE.

**Immunofluorescence assay**. A549, Calu-3, Caco-2, HEK293T, or FLAG-NLRC5 stably expressing HeLa cells were plated on coverslips overnight and transfected with DNA constructs using Lipofectamine 2000. After 24 h transfection, cells were either stimulated with IFNγ (100 U/ml) (Peprotech) or treated with 100 nM of leptomycin B for various time points, then fixed with 4% PFA for 20 min at 4 °C, permeabilized with 0.05% Triton X-100 in PBS for 5 min at room temperature (RT), blocked with 5 mg/ml BSA in PBST (0.04% Tween20 in PBS) for 30 min, and

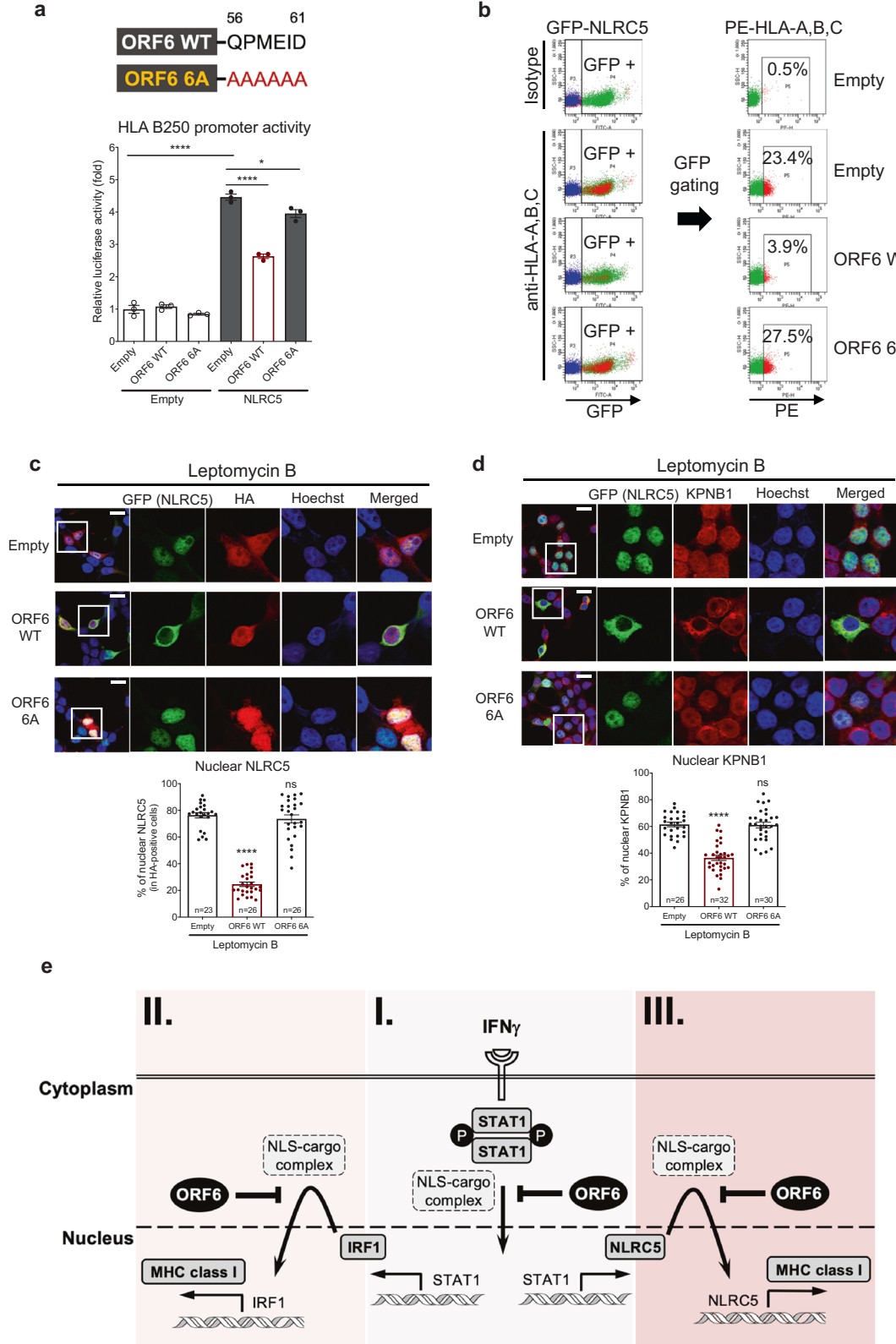

incubated with the relevant primary antibodies diluted in blocking buffer at 4 °C overnight. The cells were then incubated with secondary antibodies at room temperature for 1 h. Nuclei were stained with 1 µg/ml of Hoechst 33342 solution and analyzed with a confocal laser microscope (Olympus).

**Quantitative reverse-transcription PCR.** For the evaluation of messenger RNA expression, RNA was reverse transcribed into cDNA using ReverTra Ace™ qPCR RT Master Mix reagent (Toyobo), and quantitative reverse-transcription PCR was performed using THUNDERBIRD™ SYBR™ qPCR Mix reagent (Toyobo, Japan) with the specific primer sets targeting SARS-CoV-2 envelop[26], human *NLRC5*, *HLA-A*, *HLA-B*, *B2M*, *TAP1*, *PSMB9*, *STAT1*, *IRF1*, *IFNB*, *IL6*, *TNF*, *ISG15*, *OAS2*, and *IFI6* genes. As an internal control, the human *GAPDH* gene was targeted, amplified, and used for normalization. Relative gene expression levels were calculated, and the results are shown as fold induction over untreated control. The sequence information of the primers for RT-qPCR is shown in Supplementary Table 2.

**Fig. 7 The carboxy-terminus region of SARS-CoV-2 ORF6 is required for inhibition of the NLRC5 function. a** Role of the carboxy-terminus region of SARS-CoV-2 ORF6 on NLRC5-mediated HLA-B promoter activity. The schematic image of SARS-CoV-2 ORF6 shows the position of the mutated amino acid at ORF6 C′ terminus end. HEK293T cells were transfected with HLA-B promoter-containing reporter construct and plasmids expressing empty or NLRC5, along with plasmids expressing empty, WT, or 6 A mutant of ORF6. At 36 h after transfection, cells were collected to measure the luciferase activity. The results are from three independent experiments. **b** Effect of SARS-CoV-2 WT or 6 A mutant ORF6 on NLRC5-mediated surface expression of HLA-A, B, and C proteins analyzed by flow cytometry. GFP (NLRC5) positive cells were gated for evaluating the PE (HLAs) signal intensity (%). The FACS gating strategies are provided in Supplementary Fig. 12. **c** and **d** Immunofluorescence analysis of the cellular localization of GFP-tagged NLRC5 (**c**) or endogenous KPNB1 (**d**) by HA alone, HA-tagged SARS-CoV-2 ORF6 WT, or 6A mutant. HEK293T cells were transfected with plasmids expressing the indicated proteins. At 24 h after transfection, the cells were treated with 100 nM of Leptomycin B for 8 h, and then analyzed by confocal microscopy. Quantitative comparison of the nuclear signal intensity (% of nuclear signal intensity/total cell signal intensity) of NLRC5 or KPNB1 is shown with a bar graph using ImageJ. The scale bar indicates 20 mm. The sample numbers for evaluation are indicated in the bar graph. **e** The type II IFN system is a master host immune response for the MHC class I-mediated antigen-presenting pathway upon invasion by intracellular pathogens or cancer. I Activation of IFNGR by IFNγ triggers immediate phosphorylation of STAT1. Subsequently, phosphorylated STAT1 undergoes homodimerization, and the dimerized STAT1 complex can translocate to the nucleus. Nuclear-localized STAT1 initiates its function as a transcription factor for the expression of the IFNγ-inducible genes, including IRF1 and NLRC5. Upon production, IRF1 (II) and NLRC5 (III) translocate to the nucleus and function as transcription factors for induction of the MHC class I pathway. All three transcriptional regulators possess NLS and utilize the karyopherin-associated import complex for entering the nucleus. However, SARS-CoV-2 ORF6 targets this nuclear translocation by blocking karyopherin-mediated protein import of these MHC class I-activating transcription factors, thereby resulting in suppressed MHC class I expression upon viral infection.

**Reporter assay**. For luciferase assays, $1 \times 10^4$ of HEK293T or Calu-3 cells were seeded into 48 well plates. For analyzing the inhibitory effect of SARS-CoV-2 viral proteins on type II IFN signaling pathway, 50 ng reporter constructs of either IRF1 $3 \times$ GAS, NLRC5 promoter, or HLA promoters (described in "Methods" at 'Plasmids' section) were cotransfected with SARS-CoV-2 viral protein expression vectors, HA-Nsp1, HA-Nsp15, HA-ORF6, HA-ORF8, or HA-N. After 24 h incubation, cells were treated with IFNγ (100 U/ml) for 9 h. For analyzing the inhibitory effect of SARS-CoV-2 viral proteins on IRF1- or NLRC5-mediated MHC class I expression, 50 ng of HLA-A, HLA-B, and HLA-C reporters were cotransfected with 100 ng of either GFP alone, GFP-NLRC5 wild type, or GFP-NLRC5 Walker A mutant. After 36 h incubation, cells were harvested, and cell lysates were analyzed using a Luciferase Assay System (Promega).

**RNA interference**. The siRNA negative control and siRNAs targeting human KPNB1 (sense; CUGCAUAUGAAUCUCUGAU, anti-sense; AUCAGAGAUU-CAUAUGCAG) and human KPNA6 (sense; GUGUGAUCACAAGAGAGAU, anti-sense; AUCUCUCUUGUGAUCACAC) were purchased (Bioneer). For knockdown of target genes, siRNAs were transfected into the cells with Lipofectamine 2000 (Invitrogen) according to the manufacturer's recommendations. At 48 h post-transfection, cells were used for further experiments.

**Statistics**. All non-RNA-seq statistical analyses were performed by a Two-tailed unpaired $t$-test. The result significance was indicated as *$P < 0.05$; **$P < 0.01$; ***$P < 0.001$; ****$P < 0.0001$; ns, not significant using data from three independent experiments. The error bars represent mean values ± SD. The exact $p$-values are available in the Source Data. The graphs were generated using GraphPad Prism 9 software (GraphPad Software, USA).

**Reporting summary**. Further information on research design is available in the Nature Research Reporting Summary linked to this article.

## Code availability
Scripts used to perform processing and generate figures can be found at https://github.com/scho75/RNA-seq-data-mining-analysis.

## Data availability
The RNA-seq data used in this study are available in the NCBI Gene Expression Omnibus (GEO) Repository under accession code GSE152075 for the nasopharyngeal swab dataset and under accession code GSE147507 for the normal human bronchial epithelial cell dataset. Source data for scRNA-seq data in Supplementary Fig. 1b can be accessed on the cellxgene platform (https://cellxgene.cziscience.com/e/krasnow_lab_human_lung_cell_atlas_10x-1-remixed.cxg/). The source data underlying Figs. 1d, 2a–b, 3a–f, 4a–e, 5a–b, 5d–e, 6b–f, 7a, c, d and Supplementary Figs. 2, 3a–b, 6a–b, 7, 8, 10b, and 11a are provided as a Source data file. No additional dataset has been generated in the current study. Source data are provided with this paper.

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

## Acknowledgements
The authors thank Dr. Takashi Fujita, Dr. Peter Van den Elsen, and Dr. Shigetsugu Hatakeyama for providing reagents, and C. Matsukawa for secretarial assistance. This work was supported by grants from the American Lung Association LCD-507710, National Multiple Sclerosis Society, CSTR, TAM Genomics, Japan Society for the Promotion of Science (JSPS) Grant number: 18H06135 and 20K21511, SENSHIN Medical Research Foundation, Bristol Myers Squibb grant, Takeda Science Foundation, Kaketsu-ken foundation grant, Kobayashi Foundation grant to K.S.K., Japan Society for the Promotion of Science (JSPS) Grant number: 19K23837 and 21K15445 to J-S.Y., and the Japan Program for Infectious Diseases Research and Infrastructure (JP 20wm022500) from Japan Agency for Medical Research and Development (AMED) to H.S., Japan Society for the Promotion of Science (JSPS) Grant number: 19K16681, The Hitachi Global Foundation to R.O.

## Author contributions
J-S.Y. and K.S.K. conceived and designed the study. J-S.Y., M.S., S.X.C., Y.O., H.S., and K.S.K. designed experiments. J-S.Y., M.S., S.X.C., Y.K., B.Z., R.O., and Y.O. performed the experiments. J-S.Y., P.d.F., H.S., and K.S.K. performed data analysis. J-S.Y., S.X.C., and K.S.K. wrote the manuscript.

## Competing interests
The authors declare no competing interests.
