## [Peer Review File · Nature Communications]

SARS-CoV-2 Inhibits Induction of the MHC Class I Pathway by Targeting the STAT1-IRF1-NLRC5 AxisREVIEWER COMMENTS

Reviewer #1 (Remarks to the Author):

In this study the authors claim to demonstrate that SARS-CoV-2 targets key MHC class I transcriptional regulators, STAT1-IRF1-NLRC5. They claim that MHC induction is inhibited by SARS-CoV-2 infection.

Much of the work in this manuscript reiterates published findings, and the observations that are novel are at best incremental and quite marginal advancements over the existing state of knowledge.

For example, it has already been described that SARS-CoV-2 (and SARS-CoV-1) ORF6 blocks nuclear translocation of STAT1, STAT2 and IRF3 in response to type I IFN, by counteracting cellular nuclear import pathways. The descriptions in this manuscript that SARS-CoV-2 ORF6 blocks nuclear translocation of STAT 1 in response to type II IFN, and that their own transcription factor of interest, NLRC5 is also blocked, is therefore of limited interest, being predictable from the existing state of knowledge.

There are observations that will be of interest to some, but unfortunately I do not believe they are of sufficient impact to justify publication in an inter-disciplinary journal such as Nature Communications.

While for the most part, the work appears to have been competently conducted, I find some issues which I would very much hope the authors will consider before their next submission. These mainly relate to the attempts by the authors to imply that SARS-CoV-2 is specifically targeting the MHC-I pathway rather than IFN signalling overall, especially in the first figure.

1) The analysis in figure one of existing datasets is extremely selective and could be described as misleading. I will outline my problems with these datasets but overall my advice would be that the authors remove this figure, and the associated claims in future submissions.

The authors segregate the COVID19 patients from the Lieberman et al by the level of CD45 expression, to exclude patients with high levels of immune infiltrates, as they wish to assay the response in epithelial cells. This may be appropriate, but from the data presented, it is impossible to tell if the comparison with healthy donors is at all valid. The most important thing would be a comparison of the level of CD45 between the healthy donors and 'low immune flux' patients. Clearly if CD45 levels were lower in the low immune flux patients than the healthy donors then all of this analysis is invalid. This may be the case looking at the heat-map, as by virtue of the analysis all of the 'lo flux' patients have low CD45 whereas the negative patients includes a mixture of CD45 levels. That being said, I may be wrong, as the heat-map is essentially unintelligible as no information is given as what values are being depicted. Presumably the scale shows log2 fold change values but the denominator is not given.

It would be important for the authors to include other IFN markers to show whether the effect is really specific to MHC class I or if it is a general IFN response.

The authors claim that the significant differences they see are due to the changes in SARS-CoV-2 infected cells. But the authors must consider the number of infected cells in the samples from infected patients is clearly not 100% - indeed, the infected cells are likely to make up a relatively small proportion. The cell-autonomous mechanism by which infected SARS-CoV-2 infected cells prevent MHC-I up-regulation is therefore unlikely to account for the changes described – in fact, it makes me feel that this is even more likely to be an artefact of their subgroup analysis.

In their analysis of the data from Blanco-Mello et al the authors show only MHC class I related genes, but Blanco-Mello et al clearly show that SARS-CoV-2 infection at low MOI fails to induce a broad class I interferon response. A very similar graph as those shown could be generated for any of the IFN-inducible genes, and thus this panel is misleading in the same way as 1C.

2) The bottom panel of Figure 2C is rather odd. The data appears to have been collected on a linear, as opposed to logarithmic scale, and yet the differences when cells are stimulated are extremely modest such that the measurements presented here are in doubt.

3) It will be of interest that the authors were unable to replicate the pre-print observations regarding MHC-I downregulation by ORF8, however, the experiments have not been conducted or presented in such a way as to demonstrate that this is clearly the case. The authors only ever seem to test the expression of class I when ORF8 is co-transfected with NLRC5 which makes the results difficult to interpret. It is worth noting that the authors suggest the reason for the difference between this manuscript and the preprint by Zhang et al is that the effect may be cell type specific – but both groups use the same cell line - 293T cells.

4) The authors repeatedly state without any reference that IFN-gamma (and not type I IFN) is the primary regulator of class I expression or that IFN-gamma is more potent. Please include the appropriate references for this statement in all cases.

Reviewer #2 (Remarks to the Author):

The authors show that SARS-CoV-2 downregulates expression of MHC class I both in infected epithelial cells and COVID-19 patients. They report that the ORF6 protein suppresses IFN-gamma-mediated STAT1 signaling, with a reduced expression of BLRC5 and IRF1, and also targets the nuclear import of NLRC5 mediated by the karyopherin complex.

The findings reported are of high relevance to understand immune evasion by SARS-CoV-2. The authors present a vast amount of experimental work to dissect how SARS-CoV-2 infection controls several pathways and to demonstrate the implication of ORF6. The experimental data support the conclusions.

Specific comments:

Fig.2b. The authors indicate that 'induction of gene expression of the primary transcriptional regulators responsible for the MHC class I activation, such as NLRC5, IRF1 and STAT1 was also remarkably suppressed by SARS-CoV-2 infection'. But the results shown in Fig. 2b for STAT1 expression show no statistical significance in infected Calu-3 cells (Fig. 2b), nor a reduction of STAT1 is observed in Caco-2 or Huh7 cells (Suppl. Fig. 2). The authors do find downregulation of MHC class I protein expression by flow cytometry (Fig. 2c), but the transcriptional data on STAT1 does not correlate with a reduction in NLRC5 and IRF1 expression (Fig. 2b and Suppl. Fig. 2). To make these conclusions the authors compare the results in infected cells with the induction observed after Poly(I:C) treatment, a very strong activator of the IFN response. It is not clear whether this is the correct comparison, the control demonstrates that the cells can activate the IFN response, but IFN activation in the context of infection may not be that potent (even in the absence of viral inhibitors of these pathways). The ideal control would be the infection with a SARS-CoV-2 mutant lacking the ORF6 gene. The authors should clarify this.

Fig. 3b. The expression of IFNB gene is not suppressed after SARS-CoV-2 infection. Again, the authors may be comparing the results of Poly(I:C), a very strong activator of the IFN response, with SARS-CoV-2 infections. Whichever the comparison, in this case, there is no inhibition of the IFNB gene expression in infected cells.

Fig. 4a. The results show that expression from the IRF1 promoter is also reduced, to a lesser extent compared to ORF6, in the presence of Nsp1 and ORF8 ... the authors should comment on this in the Results section. A similar effect is seen with Nsp1 when testing the NLRC5 promoter activity. The Discussion should include alternative roles of Nsp1 and ORF8.

Discussion, page 15. It is interesting that impaired expression/function of NLRC5 may influence the levels of MHC class I and affect the CD8 response in cancer patients. This means that the immune response to SARS-CoV-2 and the development of COVID19 disease may also depend on the genetic variability of different patients leading to differential expression of NLRC5. The authors may comment

on this interesting possibility in the Discussion.

There are some errors with the format of the manuscript. Pages and lines are not numbered and this makes difficult the writing of the report. Some parts of Figures 3 and 6 are missing.

Reviewer #3 (Remarks to the Author):

This study by Yoo et al sets out to define the mechanism by which SARS-CoV-2 can inhibit the MHC I pathway to provide a possible explanation of why anti-viral CD8+ T cell responses in COVID-19 patients are impaired. RNA-Seq analysis of samples from COVID-19 patients as well as infected primary lung epithelial cells were performed to establish that MHC I gene expression is reduced and associated with decreased NLRC5 expression. This result was also observed using the epithelial cell line, Calu-3, where upon infection led to reduction of STAT-1 and IRF-1 expression. The authors then turn to the use of transfection system using HEK293T and HeLa cells to comprehensively define the mechanisms associated with these outcomes. Through this, they identified the ORF6 viral protein as the key player in inhibiting STAT-1 activity, IFN-g induced outcomes resulting in reduced MHC I expression at the gene and protein level, and confirmed the role of NLRC5. Additionally, experiments also defined how ORF6 inhibits NLRC5 function by blocking karyopherin complex-mediated nuclear import. Altogether, these findings provide an insight into a possible mechanism by which SARS-CoV-2 can evade host detection. This is a comprehensive, well written and presented study.

The role of NLRC5 as a key regulator of the MHC I pathway in cancer and other viral infections is well studied and its mechanisms of action in this arena have been well defined. Thus, its role during SARS-CoV-2 infection is perhaps not completely unexpected. Nonetheless, this work serves to provide first evidence of its possible involvement in this setting, particularly in infected human samples and notably epithelial cells. Key components of the pathway that are affected are unravelled and the role of ORF6 is clearly implicated, albeit this is studied in non-respiratory continuous cell lines. While the reviewer understands that these systems lend themselves well for such mechanistic studies and are also more accessible, confirming some of the later major findings, especially those relating to ORF6 mediated NLRC5 function and karyopherin-mediated nuclear import, in respiratory epithelial cells are encouraged. These data would strengthen the conclusions of the manuscript.

Other comments:

Fig. 1a Data on MHC-I downregulation in SARS-CoV-2 patients in comparison to healthy controls should be also verified at the protein level by flow cytometry.

Supplementary Figure 1b. UMAP analysis of cells from normal lung tissue showed MHC I and CD45 gene expression associated with different cell populations. Given that correlation data for NLRC5 expression is available (in Supp Fig 1a), please also show the distribution of NLRC5 expression within the epithelial cell population.

Fig 2c. Validation of surface expression of MHC I in ACE-2-HEK293T cells shows that infection impairs upregulation (compared to immunostimulant treated cells). The authors posit that this also occurs in Calu-3 cells, however, it appears that upregulation by immunostimulants is very minimal thus in comparison with infection, is not as convincing. Data should be plotted to show statistically significant results. Although these expression levels may be intrinsic to the cell type, another form of measure to demonstrate clearer differences, including replicates, would be useful to prove that the effects observed at the gene expression level are also observed at the protein level (i.e. by Western blot).

Supplementary Figure 2 title: States that it is "epithelial cells" but the results are using Caco-2 and Huh7 which are colon and liver cells.

Fig 3b. In Calu-3 cells, elements of viral and sensing pathways are impaired by infection. Of note,

IFN- β and OAS2 were not affected which implies that inhibition targets predominantly Type II rather than Type I pathways. However, given that there are many other studies which have shown that Type I IFN pathways are impaired by SARs-CoV2 infection (including in Calu-3 cells), can the authors please provide an explanation for this? It would also be helpful if Fig 3b can be enlarged.

Fig 3e. In HEK293T cells, IFN-g-mediated STAT gene expression is inhibited by ORF expression. This is only shown for ORF6 and results of how the other viral proteins affect STAT-1 gene expression would add weight to the conclusion that that they do not have an effect on STAT signalling (in Fig 3c and d). The x-axis labels in my version are also missing and the bottom of Fig 3f is cut-off.

Fig 4a and b. In HEK293T cells, IFN-g induced IRF1 GAS and NLRC5 promoter activity is reduced by overexpression of ORF6. Graphs for depicting promoter activity (%) do not appear to show error bars, suggesting this is a single replicate. If the results are from 3 independently conducted experiments, individual replicates should be included to present the range of variation for each group.

Fig 5c. Gate frequencies in dot plots are ineligible.
In Figure legend for Fig5c, "effect of SARS-CoV-2 proteins on NLRC5-mediated on the combined surface expression...." Please clarify what is being mediated by NLRC5.

Fig 5d. In HeLa cells, NLRC5 stably expressed cells were treated with Leptomycin B where ORF6 expression showed reduced NLRC5 in the nucleus. My understanding is that Leptomycin B inhibits the export of proteins from the nucleus into the cytoplasm and yet in this experiment, what is measured is the translocation of NLRC5 into the nucleus. Can the authors please clarify the mechanisms/process of how Leptomycin B is affecting the presence of NLRC5 in the nucleus?

Fig 5e. Shown in the bar graph are only results from one experiment (with no error bars). Please present results incorporating variation between experiments or replicates. The bottom of gel image in Fig 5e is cut-off.

Reviewer #1 (Remarks to the Author):

Q: While for the most part, the work appears to have been competently conducted, I find some issues which I would very much hope the authors will consider before their next submission. These mainly relate to the attempts by the authors to imply that SARS-CoV-2 is specifically targeting the MHC-I pathway rather than IFN signalling overall, especially in the first figure.

A: We thank this reviewer for his/her appreciation that the study has been competently conducted. While it may have appeared so in the original version of the manuscript, we are not attempting to imply that SARS-CoV-2 specifically targets the MHC-I pathway rather than IFN signaling. Indeed, our data in this manuscript and data from other groups show that SARS-CoV-2 has successfully evolved to target BOTH IFN signaling as well as the MHC-I pathway. In order to avoid misunderstanding, we repeatedly emphasize this notion in the revised manuscript. (page 5 line 104, page 8 line 184 (section for Fig. 3), and page 9 line 216 (section for Fig. 4)).

Q: The analysis in figure one of existing datasets is extremely selective and could be described as misleading. I will outline my problems with these datasets but overall my advice would be that the authors remove this figure, and the associated claims in future submissions.

A: We thank this reviewer for pointing out the potential risk of misleading the reader of Figure 1. We have reanalyzed the data based on this reviewer's suggestion as illustrated below. The newly created Figure 1 is more convincing with much less risk of misleading the reader. While we think the new Figure 1 is appropriate to present in this manuscript, if this and other reviewers and/or the editors think it is preferable to remove the revised Figure 1, we will comply with this request.

Q: The authors segregate the COVID19 patients from the Lieberman et al by the level of CD45 expression, to exclude patients with high levels of immune infiltrates, as they wish

to assay the response in epithelial cells. This may be appropriate, but from the data presented, it is impossible to tell if the comparison with healthy donors is at all valid. The most important thing would be a comparison of the level of CD45 between the healthy donors and 'low immune flux' patients. Clearly if CD45 levels were lower in the low immune flux patients than the healthy donors then all of this analysis is invalid. This may be the case looking at the heat-map, as by virtue of the analysis all of the 'lo flux' patients have low CD45 whereas the negative patients includes a mixture of CD45 levels.

A: We agree that the level of CD45 between the healthy donors and the patient group should be comparable in order to analyze the differences in the gene expression between the two groups. To minimize the potential influence of high immune cell influx (high CD45 expression) on MHC class I gene expression in SARS-CoV-2 infected patients, we used the expression range of CD45 in non-infected patients as inclusion criteria and generated new Figure 1. More specifically, we utilized lower bound = 0 and upper bound = 75% quartile + IQR*1.5. New Figure 1 demonstrated that although the CD45 level is still slightly elevated in the infected group (Fig. 1b), the expression of MHC class I (HLA-A, -B and -C) and related genes (B2M and TAP1) was not increased. Since infiltrated immune cells tend to express high MHC class I genes (Supplementary Fig 1a and b), this data more clearly indicates that MHC class I was not induced in SARS-CoV-2 infected airway epithelial cells. This also fits with the independent data in Fig 1d and supplementary Fig 2 for MHC class I expression in bronchial epithelial cells upon ex vivo SARS-CoV-2 infection.

Q: That being said, I may be wrong, as the heat-map is essentially unintelligible as no information is given as what values are being depicted. Presumably the scale shows log2 fold change values but the denominator is not given.

A: We apologize for this deficiency, which was caused by a formatting error. The value is z-score transformed log2 normalized count. The information is included in the figure legend. The scale and denominator for each value have been added to Figure 1.

Q: It would be important for the authors to include other IFN markers to show whether the

effect is really specific to MHC class I or if it is a general IFN response.

A: We agree it is important to analyze the IFN gene expression whether the effect is really specific to MHC class I or if it is to both the IFN response and MHC class I. We analyzed the expression of IFN genes as suggested. At the end of this point-by-point response, please find attached Appendix Figure 1, which shows the expression of IFNs and IFN-related genes. As expected, the IFN gene was not expressed in epithelial cells in the SARS-CoV-2 non-infected group. IFN gene expression was not induced in most samples even upon SARS-CoV-2 infection, although sporadic elevation of IFN gene expression was observed in the SARS-CoV-2-infected group. This finding may suggest that decreases of HLA-A, -B, and -C genes in the airway epithelial cells upon SARS-CoV-2 infection may not be simply due to altered IFN gene expression. Since other data have more conclusively shown that the immune evasion mechanism of SARS-CoV-2 includes both suppression of IFN signaling and MHC class I signaling, it may not be worthwhile to add this Appendix Figure 1 in the manuscript. However, if this or other reviewers think it may be preferable, we will include this figure.

Q: The authors claim that the significant differences they see are due to the changes in SARS-CoV-2 infected cells. But the authors must consider the number of infected cells in the samples from infected patients is clearly not 100% - indeed, the infected cells are likely to make up a relatively small proportion. The cell-autonomous mechanism by which infected SARS-CoV-2 infected cells prevent MHC-I up-regulation is therefore unlikely to account for the changes described – in fact, it makes me feel that this is even more likely to be an artefact of their subgroup analysis.

A: We agree that the number of infected cells in the samples from infected patients is not 100%. In order to distinguish the gene expression between the infected cells and non-infected airway epithelial cells, we used an independent single-cell RNAseq data set where the gene expression in both infected and bystander ciliated bronchial epithelial cells was included. Please find new Supplementary Figure 2. The data are presented as the Log2 fold changes in the gene expression in infected cells over non-infected bystander cells. In addition to the MHC class I genes, the expression of cytokine and IFN genes has been plotted as controls. We found that MHC class I gene

expression was not induced upon SARS-CoV-2 infection, even in this dataset, further supporting the failure of induction of the MHC class I genes in SARS-CoV-2 infected airway epithelial cells. We thank this reviewer for pointing out this critical point.

Q: In their analysis of the data from Blanco-Mello et al the authors show only MHC class I related genes, but Blanco-Mello et al clearly show that SARS-CoV-2 infection at low MOI fails to induce a broad class I interferon response. A very similar graph as those shown could be generated for any of the IFN-inducible genes, and thus this panel is misleading in the same way as 1C.

A: We thank this reviewer for his/her suggestion to analyze the data for type I IFN response genes and generate a figure similar style to Figure 1d. We performed the analysis and generated such a figure (Please see Appendix Fig. 2, located at the end of this point-by-point response.). Type I IFN induction was suppressed in SARS-CoV-2 infected bronchial epithelial cells as Blanco-Mello et al showed. Since the data have been published in the original paper, although in a different format, we would like to exclude this Figure from the manuscript. However, if this or other reviewers suggest that it is preferable to present the data shown in Appendix Fig.2 in the manuscript, then we will include this figure there.

Q: The bottom panel of Figure 2C is rather odd. The data appears to have been collected on a linear, as opposed to logarithmic scale, and yet the differences when cells are stimulated are extremely modest such that the measurements presented here are in doubt

A: Lung adenocarcinoma cell line, Calu-3 expresses very modest levels of MHC class I. Therefore, we used a linear scale in order to demonstrate the difference in surface expression of MHC class I between stimulated and unstimulated groups. In order to confirm this observation using a more robust system, we have further analyzed Caco-2 (intestinal epithelial cell; new Fig 2c) and ACE2-transfected HEK293T cells (kidney epithelial cell, now moved to supplementary Fig 4). Additional data have also shown that surface expression of MHC class I was induced by stimulation as positive controls (Zika virus, Tula virus, poly(I:C), 5'-ppp RNA, and IFN γ) but not by SARS-CoV-2

infection. With additional panels, the data have become more solid. We appreciate this reviewer's suggestion.

Q: It will be of interest that the authors were unable to replicate the pre-print observations regarding MHC-I downregulation by ORF8, however, the experiments have not been conducted or presented in such a way as to demonstrate that this is clearly the case. The authors only ever seem to test the expression of class I when ORF8 is co-transfected with NLRC5 which makes the results difficult to interpret. It is worth noting that the authors suggest the reason for the difference between this manuscript and the preprint by Zhang et al is that the effect may be cell type specific – but both groups use the same cell line - 293T cells.

A: We appreciate this reviewer's pointing out that our experiments regarding to the MHC class I downregulation by ORF8 was not sufficient in order to replicate the observation in the pre-print paper. We performed the experiments in which effects of ORF8 and ORF6 on basal levels of MHC class I expression under steady-state conditions without NLRC5 transfection were examined. Please find Supplementary Fig. 8. ORF8 did not downregulate the expression of MHC class I in 293T cells. We performed 4 independent experiments to verify the results. We were unable to reproduce the data presented by Zhang et. al in the pre-print manuscript which has not been officially published yet. We do not know the exact reason why this group observed downregulation of MHC class I upon ORF8 transfection. The text has been modified accordingly (page 12, line 275-279).

Q: The authors repeatedly state without any reference that IFN-gamma (and not type I IFN) is the primary regulator of class I expression or that IFN-gamma is more potent. Please include the appropriate references for this statement in all cases.

A: We thank this reviewer for pointing out this deficiency. The references have been added. (page 3 line 71 and page 17 line 404)

Reviewer #2 (Remarks to the Author):

Q: The findings reported are of high relevance to understand immune evasion by SARS-CoV-2. The authors present a vast amount of experimental work to dissect how SARS-CoV-2 infection controls several pathways and to demonstrate the implication of ORF6. The experimental data support the conclusions.

A: We thank this reviewer for his/her appreciation of the high relevance of the study, the vast amount of experimental work, and the experimental data to support the conclusion.

Q: Fig.2b. The authors indicate that ‘induction of gene expression of the primary transcriptional regulators responsible for the MHC class I activation, such as NLRC5, IRF1 and STAT1 was also remarkably suppressed by SARS-CoV-2 infection’. But the results shown in Fig. 2b for STAT1 expression show no statistical significance in infected Calu-3 cells (Fig. 2b), nor a reduction of STAT1 is observed in Caco-2 or Huh7 cells (Suppl. Fig. 2).

A: We apologize that our description of the data was not accurate. These transcriptional regulators were not induced upon SARS-CoV-2 infection in Calu-3 and Huh-7 cells. In the Caco-2 cell, the expression of STAT1 and NLRC5 was induced, although at a very modest level. We modified the text in the Results section (page 7, line 166-168) to reflect these corrections.

Q: The authors do find downregulation of MHC class I protein expression by flow cytometry (Fig. 2c), but the transcriptional data on STAT1 does not correlate with a reduction in NLRC5 and IRF1 expression (Fig. 2b and Suppl. Fig. 2).

A: This reviewer correctly pointed out that the transcriptional data on STAT1 does not correlate with a reduction in NLRC5 and IRF1 expression. Although the exact mechanism of the discrepancy of the expression of STAT1 and NLRC5/IRF1 is not clear, it might be possible that inhibition of the function of STAT1 by ORF6 might play a role. Although the STAT1 expression level was not down-regulated by SARS-CoV-2 infection, the function of STAT1 (judging from STAT1 nuclear translocation) is blocked by SARS-

CoV-2 ORF6 (Fig. 3d). This may result in the transcriptional suppression of IRF1 and NLRC5 gene expression, both of which play a role in the transcription of components in the MHC class I pathway.

Q: To make these conclusions the authors compare the results in infected cells with the induction observed after Poly(I:C) treatment, a very strong activator of the IFN response. It is not clear whether this is the correct comparison, the control demonstrates that the cells can activate the IFN response, but IFN activation in the context of infection may not be that potent (even in the absence of viral inhibitors of these pathways). The ideal control would be the infection with a SARS-CoV-2 mutant lacking the ORF6 gene. The authors should clarify this.

A: We agree that poly(I:C) treatment is a strong activator of the IFN response and may not be an appropriate positive control to compare with SARS-CoV-2 infection. We also appreciate the excellent suggestion of using mutant SARS-CoV-2 lacking the ORF6 gene. While it is intriguing, mutant SARS-CoV-2 lacking ORF6 may not be a mere control, and substantial effort should be paid for the creation and characterization since the phenotype of ORF6 mutant virus may not be restricted to the expression of MHC class I. Therefore, we would like to reserve the construction and use of an ORF6 mutant SARS-CoV-2 for a future project. Instead, we have used other viral strains as comparisons. We chose Zika virus (*Flaviviridae*) and Tula virus (*Hantaviridae*) because these viruses are known to inhibit the interferon signaling pathways similar to SARS-CoV-2. Please find new Fig. 2c, Fig. 3b, and supplementary Fig. 5. SARS-CoV-2 infection failed to induce MHC class I in Calu-3 (lung epithelial cell line), Caco-2 (intestinal epithelial cell line), and ACE2-transfected HEK293T cell (kidney epithelial cell line), while infection with ZIKV and TULV induced MHC class I.

Q: Fig. 3b. The expression of IFNB gene is not suppressed after SARS-CoV-2 infection. Again, the authors may be comparing the results of Poly(I:C), a very strong activator of the IFN response, with SARS-CoV-2 infections. Whichever the comparison, in this case, there is no inhibition of the IFNB gene expression in infected cells.

A: We thank this reviewer for careful interpretation of the IFNB gene data and for pointing out that the expression of IFNB gene is not suppressed after SARS-CoV-2 infection. It has been reported that 'SARS-CoV-2 induces substantial but delayed IFN- β production' (Lei et al., Nature Communications, 2020, 11, 3810, doi:10.1038/s41467-020-17665-9 and Yin et al., Cell Reports, 2021, 34, 108628). Our IFNB data was consistent with results describing delayed IFNB induction at a late phase of infection (24hpi) as the other group showed. We modified the text accordingly. (page 8, line 189-190)

Q: Fig. 4a. The results show that expression from the IRF1 promoter is also reduced, to a lesser extent compared to ORF6, in the presence of Nsp1 and ORF8 ... the authors should comment on this in the Results section. A similar effect is seen with Nsp1 when testing the NLRC5 promoter activity. The Discussion should include alternative roles of Nsp1 and ORF8.

A: We agree that we should comment on the effect of Nsp1 and ORF8 on the MHC class I pathway. We observed marginal reduction of IFN γ -mediated IRF1 and NLRC5 promoter activity by Nsp1 and ORF8. However, only ORF8 showed an inhibitory effect on NLRC5 expression upon IFN γ stimulation. This effect was much milder compared to ORF6 (Fig. 4abc). Indeed, ORF8 did not show an inhibitory effect on surface expression of MHC class I, both under steady-state and NLRC5-overexpressing conditions (Fig. 5c and new supplementary Fig. 8). We modified the text in the Results (page 10 line 226-229, page 10 line 243-244, and page 12 line 275-279) with a short discussion about alternative roles.

Q: Discussion, page 15. It is interesting that impaired expression/function of NLRC5 may influence the levels of MHC class I and affect the CD8 response in cancer patients. This means that the immune response to SARS-CoV-2 and the development of COVID19 disease may also depend on the genetic variability of different patients leading to differential expression of NLRC5. The authors may comment on this interesting possibility in the Discussion.

A: This is an interesting possibility. We thank this reviewer for this suggestion. We have inserted a short discussion for the possible genetic variability for NLRC5 expression and subsequent susceptibility to COVID19 (page 16 line 397 to page 17 line 402).

Q: There are some errors with the format of the manuscript. Pages and lines are not numbered and this makes difficult the writing of the report. Some parts of Figures 3 and 6 are missing.

A: We apologize for any inconvenience caused by this deficiency. It seems that formatting errors were created during conversion of the files to PDF format. We have corrected these errors.

Reviewer #3 (Remarks to the Author):

Q: Altogether, these findings provide an insight into a possible mechanism by which SARS-CoV-2 can evade host detection. This is a comprehensive, well written and presented study.

A: We thank this reviewer for his/her appreciation of the study for its comprehensiveness.

Q: Key components of the pathway that are affected are unravelled and the role of ORF6 is clearly implicated, albeit this is studied in non-respiratory continuous cell lines. While the reviewer understands that these systems lend themselves well for such mechanistic studies and are also more accessible, confirming some of the later major findings, especially those relating to ORF6 mediated NLRPC5 function and karyopherin-mediated nuclear import, in respiratory epithelial cells are encouraged. These data would strengthen the conclusions of the manuscript.

A: We thank this reviewer for the suggestion to use more relevant cells for the study. In the assay for karyopherin-mediated nuclear localization of NLRC5, we used

additional cell lines including A549 (lung epithelial cell line), Calu-3 (lung epithelial cell line) and Caco-2 (intestinal epithelial cell line) (Supplementary Fig. 9). Also, we confirmed ORF6 inhibition of NLRC5 CITA function by reporter assay using the Calu-3 cell line (Supplementary Fig. 7). The Results section has been modified accordingly (page 11 line 261-262 and page 12 line 291-292). New data revealed that ORF6 inhibits NLRC5 nuclear localization also in lung epithelial cells. We again thank the reviewer for the excellent suggestion.

Q: Other comments:

Fig. 1a Data on MHC-I downregulation in SARS-CoV-2 patients in comparison to healthy controls should be also verified at the protein level by flow cytometry.

A: We agree that downregulation of MHC class I in SARS-CoV-2 patients in comparison to healthy controls would be important to verify at the protein level by flow cytometry. Unfortunately, there would be difficulty to perform flow cytometric analysis of human primary epithelial cells due to both technical and ethical issues in sampling. Instead, we performed multiple epithelial cell lines, including Caco-2 (intestinal), Calu-3 (lung) and ACE2 transfected HEK293 T cells (kidney) to confirm the suppression of induction of MHC class I on cell surface upon SARS-CoV-2. We also used additional viral strains, Zika and Tula viruses as controls to verify the results (Fig. 2c). Overall, the dataset to indicate the suppression of induction of MHC class I by SARS-CoV-2 are more solid in the revised manuscript.

Q: Supplementary Figure 1b. UMAP analysis of cells from normal lung tissue showed MHC I and CD45 gene expression associated with different cell populations. Given that correlation data for NLRC5 expression is available (in Supp Fig 1a), please also show the distribution of NLRC5 expression within the epithelial cell population.

A: We thank for this reviewer's suggestion. First, just in case there was a misunderstanding, the data in Supplementary Fig. 1a and 1b are from independent datasets. Nevertheless, we have added the distribution of NLRC5 expression in Supplementary Fig. 1b.

Q: Fig 2c. Validation of surface expression of MHC I in ACE-2-HEK293T cells shows that infection impairs upregulation (compared to immunostimulant treated cells). The authors posit that this also occurs in Calu-3 cells, however, it appears that upregulation by immunostimulants is very minimal thus in comparison with infection, is not as convincing. Data should be plotted to show statistically significant results. Although these expression levels may be intrinsic to the cell type, another form of measure to demonstrate clearer differences, including replicates, would be useful to prove that the effects observed at the gene expression level are also observed at the protein level (i.e. by Western blot).

A: We agree that the upregulation of MHC class I in Calu-3 by immunostimulants in the original figure, compared to those in ACE2-transfected HEK293T cells (new supplementary Fig. 4), was very minimal. The lung adenocarcinoma cell line, Calu-3, expresses very modest levels of MHC class I, even after stimulation. In order to confirm the observation in Calu-3 cells using a more robust system, we have analyzed Caco-2 (intestinal epithelial cell; new Fig 2c) cells. Additional data have also shown that surface expression of MHC class I was induced by stimulation as positive controls (ZIKA and Tula viruses, and IFN γ) but not by SARS-CoV-2 infection. With additional panels, the data have become more solid. We appreciate this reviewer for pointing out this weakness.

Q: Supplementary Figure 2 title: States that it is “epithelial cells” but the results are using Caco-2 and Huh7 which are colon and liver cells.

A: We thank this reviewer for pointing out this issue. We modified the title of Supplementary Fig. 2 as follows: “SARS-CoV-2 inhibits the upregulation of genes in the MHC class I pathway in the colon and liver carcinoma epithelial cells.”

Q: Fig 3b. In Calu-3 cells, elements of viral and sensing pathways are impaired by infection. Of note, IFN-b and OAS2 were not affected which implies that inhibition targets predominantly Type II rather than Type I pathways. However, given that there are many other studies which have shown that Type I IFN pathways are impaired by SARs-CoV2 infection (including in Calu-3 cells), can the authors please provide an explanation for this?

A: We thank this reviewer for careful interpretation of expression data for *IFN- β* and *OAS2* and pointing out that the expression of these genes was not suppressed after SARS-CoV-2 infection. It has been reported that ‘SARS-CoV-2 induces substantial but delayed IFN- β production’ (Lei et al., Nature Communications, 2020, 11, 3810, doi:10.1038/s41467-020-17665-9 and Yin et al., Cell Reports, 2021, 34, 108628). Our IFNB data was consistent with results of delayed IFN β induction at a late phase (24hpi) as the other group showed. We modified the text accordingly (page 8, line 189-190).

Q: It would also be helpful if Fig 3b can be enlarged.

A: Please find revised Fig 3b which has been enlarged for better viewing. We thank the reviewer for the suggestion.

Q: Fig 3e. In HEK293T cells, IFN-g-mediated STAT gene expression is inhibited by ORF expression. This is only shown for ORF6 and results of how the other viral proteins affect STAT-1 gene expression would add weight to the conclusion that that they do not have an effect on STAT signalling (in Fig 3c and d).

A: This is an excellent point. We performed the experiments for STAT-1 gene expression in HEK293T cells expressing other SARS-CoV-2 genes. Please find Supplementary Fig. 6. Other SARS-CoV-2 genes, including Nsp1, Nsp15, ORF8 or N did not suppress the expression of STAT-1 at both steady-state and after IFN γ stimulation. These serve as excellent controls for the function of ORF6 and we would like to thank this reviewer for suggesting this revision.

Q: The x-axis labels in my version are also missing and the bottom of Fig 3f is cut-off.

A: We apologize for any inconvenience caused by this deficiency. It seems that formatting errors were created during file conversion to a PDF format. We have corrected these errors.

Q: Fig 4a and b. In HEK293T cells, IFN-g induced IRF1 GAS and NLRC5 promoter activity is reduced by overexpression of ORF6. Graphs for depicting promoter activity (%) do not appear to show error bars, suggesting this is a single replicate. If the results are from 3 independently conducted experiments, individual replicates should be included to present the range of variation for each group.

A: We appreciate this suggestion and pointing out our shortcomings. In the revised Fig. 4a and b, we have analyzed data from multiple repeats and the error bar and P-values have been appended.

Q: Fig 5c. Gate frequencies in dot plots are ineligible.

A: Again, we appreciate for pointing out our short comings. We have improved the quality of gating figure and the font for gate frequencies has been enlarged in Fig 5c.

Q: In Figure legend for Fig5c, “effect of SARS-CoV-2 proteins on NLRC5-mediated on the combined surface expression....” Please clarify what is being mediated by NLRC5.

A: We apologize for the poor description in the figure legend. We analyzed the effect of ORF6 on the MHC class I in HEK293T cells in which GFP-tagged NLRC5 expression vector was transfected in order to upregulate MHC class I. MHC class I was detected by antibodies against HLA-A, -B and -C in flow cytometry. The description of this part in the figure legend has been modified as follow: “**Fig. 5 c** Effect of SARS-CoV-2 proteins on NLRC5-mediated surface expression of HLA-A, -B, and -C in HEK293T cells analyzed by flow cytometry. GFP (NLRC5) positive cells were gated for evaluating PE (HLAs) signal intensity.”

Q: Fig 5d. In HeLa cells, NLRC5 stably expressed cells were treated with Leptomycin B where ORF6 expression showed reduced NLRC5 in the nucleus. My understanding is that Leptomycin B inhibits the export of proteins from the nucleus into the cytoplasm and yet in this experiment, what is measured in the translocation of NLRC5 into the nucleus. Can the authors please clarify the mechanisms/process of how Leptomycin B is affecting the presence of NLRC5 in the nucleus?

A: Leptomycin B, as this reviewer correctly notes, inhibits CRM1-dependent exportation of proteins from the nucleus to the cytoplasm.

Once translated, NLRC5 protein is dynamically shuttling between the nucleus and cytoplasm, and under the steady state conditions, the majority of NLRC5 protein is observed in the cytoplasm. In our experimental conditions, we monitored whether SARS-CoV-2 ORF6 inhibited nuclear translocation of NLRC5, and we used Leptomycin B to capture NLRC5 proteins that were translocated to the nucleus. Once NLRC5 translocated to the nucleus, Leptomycin B blocked the nuclear export of NLRC5. Therefore, low frequency of the nuclear localized NLRC5 means that translocation of NLRC5 protein from cytoplasm to the nucleus is inhibited.

Q: Fig 5e. Shown in the bar graph are only results from one experiment (with no error bars). Please present results incorporating variation between experiments or replicates. The bottom of gel image in Fig 5e is cut-off.

A: We appreciate this suggestion and pointing out our shortcomings. In revised Fig. 5e, we have analyzed data from multiple repeats and the error bar and P-values have been appended. The bottom of gel image in Fig 5e has been fixed.

Appendix Fig. 1

Appendix Fig.1 (related to Fig. 1a in the manuscript)

Various IFNs and IFN-inducible genes are not induced in SARS-CoV-2-infected patients.

Appendix Fig. 2

Appendix Fig.2 (related to Fig. 1d in the manuscript)

Various IFNs and IFN-inducible genes are not induced in SARS-CoV-2-infected NHBE cells.

REVIEWER COMMENTS

Reviewer #1 (Remarks to the Author):

While the authors agree with the points raised in the previous round of review, the paper itself is not substantially different and retains many of the previous problems without bringing substantial novelty and I cannot recommend it for publication.

1) As a general point, the authors seem to have decided that when comparing different datasets or their own results, 'significantly down-regulated' and 'not changing' can be taken to be equivalent - both being 'inhibition of up-regulation'.

This has a certain logic - i.e. assume that all virus infections will induce class-I expression unless inhibited - but this also glosses over some significant issues with consistency between the data and analysis presented here. MHC-I not changing at all is repeatedly used to corroborate data where MHC-I is strongly down-regulated - with no attempt to explain why the data should be discordant.

It also leads to some fairly odd statements like: "Supplementary Fig. 4. SARS-CoV-2 inhibits surface expression of the MHC class I proteins" - to title a figure where SARS-CoV-2 does not, in fact, change MHC-I protein expression on the cell surface.

2) The first example of this is in figure 1, which retains the most significant problems of the first round of review (despite the authors claims to the contrary). In the RNA-seq analysis, the authors claim "up to 66% reduction in mean expression of MHC class I genes". The proposed mechanism for this reduction is cell-autonomous - i.e. SARS-CoV-2 infected cells reduce class-I RNA because of ORF6 blocking NLRC5.

If the cells were a uniform population this would mean that if SARS-CoV-2 infection completely reduced MHC-I RNA levels to 0, on average, 66% of cells would have to be infected in the samples. But the swab contains a mix of cells - probably 66% aren't even infectable with SARS-CoV-2, let alone infected. And as the authors will know, any the lymphocytes present contribute a disproportionately large amount of class I.

It's simply inconceivable that sufficient numbers of cells could be infected to cause this downregulation. If they wanted to confirm this they would have to look at MHC-I expression in virally infected cells – not by correlation.

I don't know what the source of this apparent down-regulation is - it cannot be related to ORF6 blocking NLRC5 in infected cells - nor can it ever provide useful evidence for this issue.

This down-regulation in the the authors' re-analysis of the patient RNA-seq data is 'supported' by the RNA-seq and scRNA data from in vitro experiments with air liquid interface cells, which shows that there is *no change* in class I expression. This is correct – but this is not the same as a significant down-regulation! Again, this is as stated e.g. by Blanco-Melo et al that there is not a substantial IFN response in those cells, not a novel observation.

3) I assume that the reason the authors are keen to find a precedent in figure 1 for class I down-regulation is that their own qPCR data shows a down-regulation of these genes. I am not aware of this being found in any of the many available RNA-seq and scRNA datasets and as the authors show in figure 1- other groups carrying out infections in the best possible model (air-liquid interface HBECS) find that there is no significant change, or even a slight up-regulation.

Furthermore, this down-regulation is not corroborated by their own flow cytometry data. - this is all somewhat strange. There is a strong down-regulation of class I genes in Calu3 but no effect at the cell surface - the authors claim that Calu3 express so little Class I that they have to measure on a

linear scale (which doesn't seem correct and is somewhat concerning). It certainly does not match our experience (where MHC I is readily measurable on Calu3 with w6/32) or the relative expression levels of HLA-A or HLA-B at the RNA level between e.g Calu3 and Caco2 on public databases e.g. at <https://portals.broadinstitute.org/ccle>

The down-regulation in Calu3 at the RNA level appears far more potent than in the other cell lines tested, where changes are more modest. In fact, as statistics used are generally not appropriate, I suspect many changes would not be significant if the correct test was used. See point 4.

None of the cells show a down-regulation of MHC-I at the cell surface. In fact, this might be consistent - e.g. very little change in Caco2 at the RNA level and no change at the protein level. But instead figure 2 presents a dramatic change at the RNA level in Calu cells and no change in Caco cells by flow cytometry - as if this was the same outcome. It is not and results in less confidence in both assays, not more. Again, the authors do not attempt to explain these discrepancies, only call it all 'inhibition of upregulation'.

In the end, ORF6, the supposed driver for this phenotype, has no effect on MHC-I expression at steady state levels by qPCR or flow cytometry, so the downregulation at the RNA level in infection is not explained.

4) Student T-tests are repeatedly used for analysis with multiple groups. This is not appropriate.

Reviewer #2 (Remarks to the Author):

The authors have addressed my concerns. I have no further comments.

Reviewer #3 (Remarks to the Author):

the authors addressed my concern and greatly improved the manuscript.

Reviewer #1 (Remarks to the Author):

Q: While the authors agree with the points raised in the previous round of review, the paper itself is not substantially different and retains many of the previous problems without bringing substantial novelty and I cannot recommend it for publication.

A: We thank this reviewer for reviewing our manuscript and providing constructive criticisms again. While inhibition of type I IFN has been demonstrated by other groups, the transcriptional regulation of the MHC class I during SARS-CoV-2 infection is poorly understood. It is well documented that the regulation of type I IFNs and MHC class I genes are different. Moreover, the biological functions of type I IFNs and MHC class I genes are distinct. In addition, this manuscript provides molecular mechanisms of impaired upregulation of MHC class I during SARS-CoV-2 infection. Therefore, our manuscript describes novel findings, which are not predictable from the previous literature.

As this reviewer pointed out correctly below, the published RNAseq data have not illustrated impaired upregulation of MHC class I, perhaps because the focus of those studies was on other subjects. Although Figure 1 in our manuscript is our starting point for finding the failed upregulation of MHC class I in SARS-CoV-2 infected airway cells, we would like to point out that a detailed analysis of MHC class I regulation has not been documented.

Q: 1) As a general point, the authors seem to have decided that when comparing different datasets or their own results, 'significantly down-regulated' and 'not changing' can be taken to be equivalent - both being 'inhibition of up-regulation'.

This has a certain logic - i.e. assume that all virus infections will induce class-I expression unless inhibited - but this also glosses over some significant issues with consistency between the data and analysis presented here. MHC-I not changing at all is repeatedly used to corroborate data where MHC-I is strongly down-regulated - with no attempt to explain why the data should be discordant.

A: We agree that “significantly down-regulated” and “not changing” are not completely equivalent. As this reviewer pointed out, if the virus infection leads to MHC class I upregulation, which has been documented in many virus strains, both of these two cases suggest that there is ‘inhibition of upregulation’ of the MHC class I pathway by the virus. We apologize that there were shortcomings in describing the data which appeared not to indicate the same outcomes. We have modified the text to describe the following figures (page 7 lines 157, 162, 168, and 169, page 16 line 382, page 17 line 440 and Supplementary information page 4 in the title of Figure 2 and page 7 in the title of Figure 4).

Regarding the differences (in terms of the resulting pattern of SARS-CoV-2 inhibition) between the data from the different experimental settings, as answered below, we think there are multiple mechanisms involved in inhibiting the expression of MHC class I genes by SARS-CoV-2. In addition, since the swab samples used for the RNAseq analysis contained mixed cell populations, those outcomes might come from the heterogeneous responses by distinct cell types. Moreover, the distinct cellular regulation mechanism for mRNA expression (detected by RT-qPCR) and surface protein expression (detected by FACS) could be a potential reason for the difference.

Although the outcomes might appear to be dependent on the cell type-specific phenotypes or experimental settings, all data unanimously showed that the upregulation of the MHC class I was NOT observed, indicating that there is a clear inhibitory mechanism possessed by SARS-CoV-2. We have modified the text to describe these findings (page 8 line 179-187).

Q: It also leads to some fairly odd statements like: "Supplementary Fig. 4. SARS-CoV-2 inhibits surface expression of the MHC class I proteins" - to title a figure where SARS-CoV-2 does not, in fact, change MHC-I protein expression on the cell surface.

A: We thank this reviewer for careful reading the text. In comparison with other control viruses, SARS-CoV-2 infection did not result in the upregulation of the surface expression of MHC class I. We have modified the text (Supplementary materials page 4 in the title of Figure 2 and page 7 in the title of Figure 4).

Q: 2) The first example of this is in figure 1, which retains the most significant problems of

the first round of review (despite the authors claims to the contrary). In the RNA-seq analysis, the authors claim "up to 66% reduction in mean expression of MHC class I genes". The proposed mechanism for this reduction is cell-autonomous - i.e. SARS-CoV-2 infected cells reduce class-I RNA because of ORF6 blocking NLRC5.

If the cells were a uniform population this would mean that if SARS-CoV-2 infection completely reduced MHC-I RNA levels to 0, on average, 66% of cells would have to be infected in the samples. But the swab contains a mix of cells - probably 66% aren't even infectable with SARS-CoV-2, let alone infected. And as the authors will know, any the lymphocytes present contribute a disproportionately large amount of class I.

A: We thank this reviewer for pointing out the interpretation of the data in Figure 1. As this reviewer correctly notes, RNAseq data was derived from experiments performed with a mixture of cell types. In this regard, we would like to emphasize that the viral infectivity and the gene expression levels in the mixed cell population are varied. For example, even in cells that were infected, the level of induction/suppression of RNA expression in each infected cell will be different. Therefore, the percentage reduction in mean mRNA expression from the mixed population cannot simply be inferred to reflect total cell infectivity.

However, lower expression of the MHC class I genes in the SARS-CoV-2 infected group may suggest possible inhibition by the virus, either directly or indirectly. Since this issue was also our own question during the study, we performed further analysis using simpler experimental settings conducted with uniform cell populations under controlled viral infection conditions, which we have shown in Figures 1d and 2.

Q: It's simply inconceivable that sufficient numbers of cells could be infected to cause this downregulation. If they wanted to confirm this they would have to look at MHC-I expression in virally infected cells – not by correlation.

A: We agree. In fact, that was exactly what we have performed for the analysis. Based on the RNA-seq data, we confirmed the 'inhibition of upregulation' of the MHC class I gene expression by SARS-CoV-2 in *ex vivo* settings (See, Figure 1d and Supplementary Figure 2 where comparison between infected vs bystander cells was performed), and *in vitro*

settings (See, Figure 2).

Q: I don't know what the source of this apparent down-regulation is - it cannot be related to ORF6 blocking NLRC5 in infected cells - nor can it ever provide useful evidence for this issue.

A: We agree. While ORF6 displays a strong inhibitory effect on the upregulation of MHC class I upon stimulated conditions, such as IFN γ treatment or NLRC5 ectopic expression, it does not seem to have much impact on the expression of the MHC class I under steady state conditions, at least in the laboratory cell lines. We think there might be multiple mechanisms involved in downregulating the MHC class I expression. Nevertheless, our data clearly showed that at least one of the immune evasion mechanisms should be 'inhibiting the upregulation' of MHC class I by SARS-CoV-2 ORF6. Other mechanisms (if any) may also cause downregulation of the MHC class I pathway. Based on this notion, we have incorporated these considerations into the discussion (page 19, lines 478-485). We thank this reviewer for carefully reading our manuscript and pointing out the effect of ORF6.

Q: This down-regulation in the authors' re-analysis of the patient RNA-seq data is 'supported' by the RNA-seq and scRNA data from in vitro experiments with air liquid interface cells, which shows that there is *no change* in class I expression. This is correct – but this is not the same as a significant down-regulation! Again, this is as stated e.g. by Blanco-Melo et al that there is not a substantial IFN response in those cells, not a novel observation.

A: Thank you for pointing out these findings. Again, as we mentioned above, these data suggest that there might be multiple mechanisms involved in the lower level of the MHC class I expression in SARS-CoV-2 infected patients. We have incorporated these considerations into the revised discussion (page 19, lines 478-485).

We appreciate that this reviewer referred to Blanco-Melo et al (Cell 2020 181:1036) paper, which provided a significant discovery in the field. Blanco-Melo et al screened gene expression in SARS-CoV-2 infected cells and demonstrated that induction of type I IFN genes was inhibited. Although they performed analysis of the type I IFN and related genes, they neither analyzed the expression of the MHC class I-related genes nor studied MHC

class I pathways. Since the transcription of type I IFNs and the MHC class I genes are differently regulated, and their functions are distinct, it is hard to conclude that the study in this manuscript is predictable from Blanco-Melo's paper. Furthermore, we have clarified the molecular mechanism of the ORF6 inhibition on CITA function of NLRC5, which has not been shown before. Thus, we believe that the study in this manuscript combines novelty with a comprehensive analysis.

Q: 3) I assume that the reason the authors are keen to find a precedent in figure 1 for class I down-regulation is that their own qPCR data shows a down-regulation of these genes. I am not aware of this being found in any of the many available RNA-seq and scRNA datasets and as the authors show in figure 1- other groups carrying out infections in the best possible model (air-liquid interface HBECS) find that there is no significant change, or even a slight up-regulation.

A: We thank this reviewer again for pointing out the "downregulation" of MHC class I gene expression which was not observed in all experimental settings. We would like to emphasize what was observed consistently through our entire study was the 'impaired upregulation' of MHC class I, not the downregulation of MHC class I genes. The failed upregulation of MHC class I was observed in all data which we analyzed, including *in vivo* gene expression in Figure 1 (*in vivo* and *ex vivo* experiments), supplementary Figure 2 (*ex vivo* experiment), and Figure 2 (*in vitro* for mRNA and surface expression level). A total 5 independent data sets unanimously indicate that upregulation of MHC class I was NOT observed during the infection of SARS-CoV-2. We modified the text accordingly (page 7 lines 157, 162, 168, and 169, page 16 line 382, page 17 line 440 and Supplementary information page 4 in the title of Figure 2 and page 7 in the title of Figure 4).

We also thank again this reviewer for pointing out the data in the best possible model - air-liquid interface human bronchial epithelial cells (HBECS). There are currently two papers describing gene expression in air-liquid interface HBECS using RNAseq or scRNAseq. One is a preprint manuscript in which the data are not accessible (bioRxiv 2020 Muray A. et.al.). The other described the gene expression in HBECS upon SARS-CoV-2 using scRNAseq under the overtime infection condition to compare the gene expression level between infected versus bystander cells (Ravindra et al, PLoS Biology, 2021). In order to address if

alterations in gene expression is cell autonomous, we analyzed this dataset comparing gene expression in infected cells versus bystander cells (Supplementary Figure 2). Please note that the increased expression of type I IFN, but not MHC class I genes, was observed in infected cells in comparison to bystander cells, strongly supporting our observation. We further performed analysis by using a uniform cell population under the controlled viral infection in order to verify the results we obtained from *in vivo* and *ex vivo* settings. Again, although there is inconsistency in a pattern of inhibition ('reduced' versus 'no change') depends on the experimental settings, all results from our analysis consistently showed an 'impaired upregulation' of the MHC class I-related genes by SARS-CoV-2 infection.

Q: Furthermore, this down-regulation is not corroborated by their own flow cytometry data. - this is all somewhat strange. There is a strong down-regulation of class I genes in Calu3 but no effect at the cell surface - the authors claim that Calu3 express so little Class I that they have to measure on a linear scale (which doesn't seem correct and is somewhat concerning). It certainly does not match our experience (where MHC I is readily measurable on Calu3 with w6/32) (we do not know his data) or the relative expression levels of HLA-A or HLA-B at the RNA level between e.g Calu3 and Caco2 on public databases e.g. at <https://portals.broadinstitute.org/ccle>

A: We agree that strong downregulation was not observed on MHC class I expression on the cell surface analyzed by flow cytometry analysis, although there was a reduction at the mRNA level in Calu-3 cell line. While we do not know the exact mechanism of these, cell surface expression level and mRNA level may not always correlate because cell surface expression is regulated not only by the transcription level but other various factors. Moreover, since HLA proteins show a relatively slow turnover rate (20hr ~ 65hr) (Prevosto et al, 2016, PLoS ONE,11(8)e0161011), the MHC class I protein level may be sustainable even if de novo synthesis of mRNA is inhibited.

Although the basal level of MHC class I in our Calu-3 cell line was very low, other experimental controls stimulated with PAMP RNAs (poly I:C or 5'ppp-RNA) or IFN γ still showed induction of the MHC class I surface expression under the same condition, indicating that the upregulation of the MHC class I surface expression is suppressed by SARS-CoV-2 infection. Furthermore, as this reviewer mentioned, we found one paper (Shilts et al,

Scientific reports, 2021 11:413), in which the authors used the anti-MHC class I antibody 'W6/32' to measure MHC class I surface expression in the Calu-3 cell line. This paper showed that although the basal level of the MHC class I was highly detectable, there was almost no induction of the MHC class I surface expression (supplementary Fig. 2, by judging the MHC class I levels in 'No sgRNA' (equivalent to 'viral infection') versus 'sgACE2' (equivalent to 'no infection') similar to our result, thereby corroborating our observation. Therefore, these observations indicate that regardless of the quality of the antibody or the inconsistent cell line status or conditions, the MHC class I is not upregulated in SARS-Cov-2 infected cells.

We cannot comment on the expression level of MHC class I in Calu3 cells observed by this reviewer or in the website that this reviewer mentioned (Cancer Cell Line Encyclopedia at the Broad Institute, <https://portals.broadinstitute.org/ccle>), both of which are not currently accessible. Since there might be differences in sub-cell lines or culture conditions, we need to compare the data side by side. However, again, it was observed unanimously that MHC class I expression was NOT induced upon SARS-CoV-2 infection in all cell lines that we analyzed, regardless of the level of MHC I class I expression. We thank this reviewer for his/her careful examination of this point.

Q: The down-regulation in Calu3 at the RNA level appears far more potent than in the other cell lines tested, where changes are more modest. In fact, as statistics used are generally not appropriate, I suspect many changes would not be significant if the correct test was used. See point 4.

A: We thank this reviewer for pointing out if the correct test was used in order to analyze for the statistical significance. We agree that this is important. As we summarized in the answer for point 4 below, we confirmed that correct methods were used for all the statistical testing.

Q: None of the cells show a down-regulation of MHC-I at the cell surface. In fact, this might be consistent - e.g. very little change in Caco2 at the RNA level and no change at the protein

level. But instead figure 2 presents a dramatic change at the RNA level in Calu cells and no change in Caco cells by flow cytometry - as if this was the same outcome. It is not and results in less confidence in both assays, not more. Again, the authors do not attempt to explain these discrepancies, only call it all 'inhibition of upregulation'.

A: We agree that although all three cell lines showed a reduced mRNA expression by SARS-CoV-2 infection, downregulation of MHC class I on the cell surface was not observed in all cell lines. As we mentioned above, cell surface expression level and mRNA level may not always correlate because cell surface expression is regulated not only by the transcription level. Since the turnover rate of HLA proteins is relatively long, it is more likely that the regulation of the HLA protein half-life may be involved in the discrepancy between mRNA level and protein level. Again, it is important to note that upregulation of surface expression of MHC class I was NOT observed in any cell lines that we used for the entire study. Therefore, these data unanimously suggest that there is a mechanism to suppress the upregulation of the MHC class I pathway at the mRNA and cell surface level. As requested by this reviewer, we modified the text to describe the difference among cell types (page 8, lines 179-187). We thank this reviewer for the careful data interpretation.

Q: In the end, ORF6, the supposed driver for this phenotype, has no effect on MHC-I expression at steady state levels by qPCR or flow cytometry, so the downregulation at the RNA level in infection is not explained.

A: We agree that ORF6 did not reduce the expression of MHC class I under the steady-state condition. As we explained above, ORF6 suppressed the function of three key molecules, STAT1, IRF1, and NLRC5 all of which are critical for the upregulation of MHC class I upon IFN γ treatment and under the inflammatory conditions. It is likely that besides ORF6, there might be alternative mechanisms to suppress the MHC class I expression at steady state. We incorporated a consideration of this possibility in the revised Discussion section (page 19, lines 478-485).

Q: 4) Student T-tests are repeatedly used for analysis with multiple groups. This is not appropriate.

A: We thank this reviewer to point out the methods of statistical testing. We are well aware of this issue that is crucial for making a scientifically sound conclusion.

As this reviewer correctly pointed out, Student T-test is NOT appropriate IF values are analyzed comparing among multiple groups. It is appropriate IF an 'unpaired t-test' is used in order to compare the mean value between two independent groups with equal variance. We used an unpaired t-test for the statistical analysis for the comparison between 'two independent groups' (e.g. 'mock' vs 'infected/stimulated' or 'empty control' vs 'viral gene expressed') as shown in Figures 3, 4, 5, 6, and 7, and Supplementary Figures 3, 5, 6, 7, 8, and 9. Therefore, we believe that the student t-test is an appropriate method to measure the statistics for these figures. Please note that the student T-test was not used for comparing the multiple groups.

However, in the case of Figure 1, since the samples comprise randomly selected values, we used the Mann-Whitney U test or Wald-test adjusted by Benjamini-Hochberg to reduce the false discovery rate.

Reviewer #2 (Remarks to the Author):

Q: The authors have addressed my concerns. I have no further comments.

A: We thank this reviewer again for acknowledging the novelty and significance of the study and providing constructive criticisms which helped to significantly improve our manuscript.

Reviewer #3 (Remarks to the Author):

Q: the authors addressed my concern and greatly improved the manuscript.

A: We thank this reviewer again for acknowledging the novelty and significance of the study and providing constructive criticisms. His/her suggestions were very helpful to improve the quality of the manuscript.